# SELF-SUPERVISED CONTRASTIVE LEARNING FOR LONG-TERM FORECASTING

**Junwoo Park, Daehoon Gwak, Jaegul Choo, Edward Choi**
Kim Jaechul Graduate School of AI, KAIST, Daejeon, Republic of Korea
{junwoo.park,daehoon.gwak,jchoo,edwardchoi}@kaist.ac.kr

## ABSTRACT

Long-term forecasting presents unique challenges due to the time and memory complexity of handling long sequences. Existing methods, which rely on sliding windows to process long sequences, struggle to effectively capture long-term variations that are partially caught within the short window (*i.e.,* outer-window variations). In this paper, we introduce a novel approach that overcomes this limitation by employing contrastive learning and enhanced decomposition architecture, specifically designed to focus on long-term variations. To this end, our contrastive loss incorporates global autocorrelation held in the whole time series, which facilitates the construction of positive and negative pairs in a self-supervised manner. When combined with our decomposition networks, our contrastive learning significantly improves long-term forecasting performance. Extensive experiments demonstrate that our approach outperforms 14 baseline models in multiple experiments over nine long-term benchmarks, especially in challenging scenarios that require a significantly long output for forecasting. Source code is available at https://github.com/junwoopark92/Self-Supervised-Contrastive-Forecsating.

## 1 INTRODUCTION

Time-series data presents a unique challenge due to its potentially infinite length accumulating over time, making it infeasible to process them all at once (Ding et al., 2015; Hyndman et al., 2015; Rakthanmanon et al., 2013). This requires different strategies compared to other sequence data such as natural language sentences. To address this, the sliding window approach (Kohzadi et al., 1996) is commonly employed to partition a single time-series data into shorter sub-sequences (*i.e.,* windows) Typically, in time-series forecasting, the sliding window approach enables models to not only process the long-time series but also capture local dependencies between the past and future sequence within the windows, resulting in accurate short-term predictions.

Recently, as the demands in the industry to predict more distant future increases (Ahmad et al., 2014; Vlahogianni et al., 2014; Zhou et al., 2021), various studies have gradually increased the window length. Transformer-based models have reduced computational costs of using long windows through improvements in the attention mechanism (Zhou et al., 2021; Wu et al., 2021; Liu et al., 2022a). Also, CNN-based models (Bai et al., 2018; Yue et al., 2022) have applied a dilation in convolution operations to learn more distant dependencies while benefiting from their efficient computational cost. Despite the remarkable progress made by these models, their effectiveness in long-term forecasting remains uncertain. Since the extended window is still shorter than the total time series length, these models may not learn the longer temporal patterns than the window length.

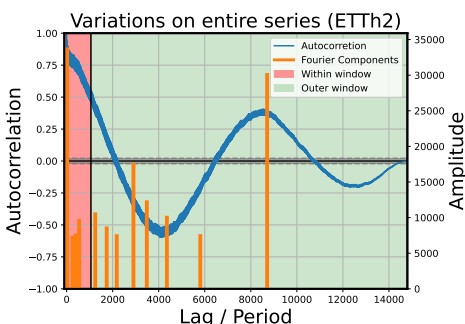

Figure 1: Long-term variations span beyond the conventional window. There are non-zero correlations (Left Y axis) with longer lags, and Fourier components (Right Y axis) with longer periods than the window size.

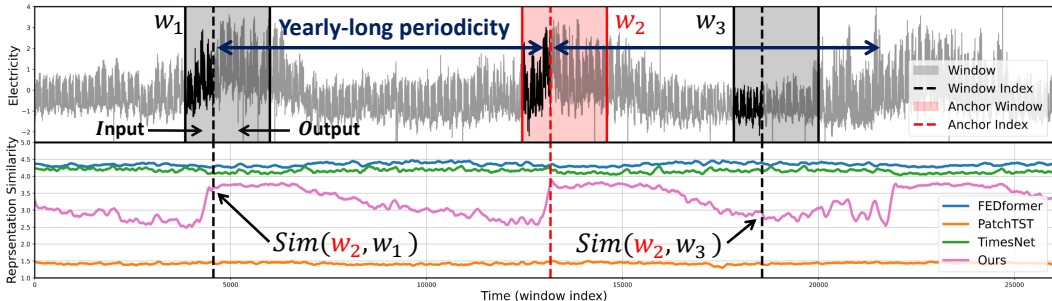

Figure 2: (Top) Electricity time series including a long-term variation beyond window size. (Bottom) Plotted representation similarities of four models between an anchor window $\mathcal{W}_2$ and all other windows including $\mathcal{W}_1$ and $\mathcal{W}_3$. To clearly highlight long-term correlation, we smoothed fluctuations caused by short-term correlation. The details of the visualization are found in Appendix C.1. Even though $\mathcal{W}_2$ have a similar temporal pattern with $\mathcal{W}_1$ due to yearly-long periodicity, three models, except for Ours, fail to learn this periodicity as the representation. The three models result in nearly identical cosine similarity scores (*i.e.,* $Sim(\mathcal{W}_2, \mathcal{W}_1) \approx Sim(\mathcal{W}_2, \mathcal{W}_3)$) between two representations of input parts within each window. This contributes to our model showing lower mean squared errors (0.275) in long-term predictions than PatchTST (0.332) and TimesNet (0.417).

In this paper, we first analyze the limitations of existing models trained with sub-sequences (*i.e.,* based on sliding windows) for long-term forecasting tasks. We observed that most time series often contain long-term variations with periods longer than conventional window lengths as shown in Figure 1 and Figure 5. If a model successfully captures these long-term variations, we expect the representations of two distant yet correlated windows to be more similar than uncorrelated ones. However, since the previous studies all treat each window independently during training, it is challenging for the model to capture such long-term variations across distinct windows. Explicitly, Figure 2 shows that the representations of existing models fail to reflect the long-term correlations between two distant windows. Still, recent methods tend to overlook long-term variations by focusing more on learning short-term variations within the window. For example, existing models based on decomposition approaches (Zeng et al., 2023; Wang et al., 2023) often treat the long-term variations partially caught in the window as simple non-periodic trends and employ a linear model to extend the past trend into the prediction. Besides, window-unit normalization methods (Kim et al., 2021; Zeng et al., 2023) can hinder long-term prediction by normalizing numerically significant values (*e.g.,* maximum, minimum, domain-specific values in the past) that may have a long-term impact on the time series. Since these normalization methods are essential for mitigating distribution shift problems (Kim et al., 2021) caused by nonstationarity (Liu et al., 2022b), a new approach is necessary to learn long-term variations while still keeping the normalization methods.

Therefore, we propose a novel contrastive learning to help the model capture long-term dependencies that exist across different windows. Our method builds on the fact that a mini-batch can consist of windows that are temporally far apart. It allows the interval between windows to span the entire series length, which is much longer than the window length. Section 3.1 describes the details of our contrastive loss. Moreover, we use our contrastive loss in combination with a decomposition-based model architecture, which consists two branches, namely a short-term branch and a long-term branch. Naturally, our loss is applied to the long-term branch. However, as pointed out earlier, the long-term branch in the existing decomposition architecture has been composed of a single linear layer, which is unsuitable for learning long-term representations. Thus, as explained in Section 3.2, we redesign the decomposition architecture where the long-term branch has sufficient capacity to learn long-term representation from our loss. In summary, the main contributions of our work are as follows:

- Our findings reveal that the long-term performances of existing models are poor as those models overlooked the long-term variations beyond the window.

- We propose AutoCon, a novel contrastive loss function to learn a long-term representation by constructing positive and negative pairs across distant windows in a self-supervised manner.

- Extensive experiments on nine datasets demonstrate that the proposed decomposition architecture trained with AutoCon achieves performance improvements of up to 34% compared to a total of 14 concurrent models including three representation methods.

## 2 RELATED WORK

**Contrastive Learning for Time-series Forecasting** Contrastive learning (Chen et al., 2020; Khosla et al., 2020; Zha et al., 2022) is a type of self-supervised learning technique that helps models learn useful representations of data without the need for explicit labeling of data. Motivated by the recent success of contrastive learning in computer vision, numerous methods (Tonekaboni et al., 2021; Yue et al., 2022; Woo et al., 2022a) have been proposed in time-series analysis. In contrastive learning, since how to construct positive pairs has a great influence on the performance, they mainly proposed positive pair construction strategies such as temporal consistency (Tonekaboni et al., 2021), sub-series consistency (Franceschi et al., 2019), and contextual consistency (Yue et al., 2022). However, these strategies have a limitation in that only temporally close samples are selected as positives, overlooking the periodicity in the time series. Due to the periodicity, there may be more similar negative samples than positively selected samples. Recently, CoST (Woo et al., 2022a) tried to learn a representation considering periodicity through Frequency Domain Contrastive loss, but it could not consider periodicity beyond the window length because it still uses augmentation for the window. In the time-series learning framework, we focus on the fact that randomly sampled sequences in a batch can be far from each other in time. Therefore, we propose a novel selection strategy to choose not only local positive pairs but also global positive pairs between the windows in the batch.

**Decomposition-based Models for Long-term Forecasting** Time-series decomposition (Cleveland et al., 1990) is a well-established technique that involves breaking down a time series into its individual components, such as trend, seasonal, and remainder components. By decomposing a time series into these components, it becomes easier to analyze each component's behavior and make more interpretable predictions. Thus, decomposition-based models (Wu et al., 2021; Zhou et al., 2022b; Wang et al., 2023) have gained popularity in time-series forecasting, as they offer robust and interpretable predictions, even when trained on complex time series. Recently, DLinear(Zeng et al., 2023) has demonstrated exceptional performance by using a decomposition block and a single linear layer for each trend and seasonal component. However, our analysis indicates that these linear models are effective in capturing high-frequency components that impact short-term predictions, while they often miss low-frequency components that significantly affect long-term predictions. Therefore, a single linear model may be sufficient for short-term prediction, but it is inadequate for long-term prediction. In light of this limitation, we propose a model architecture that includes layers with varying capacities to account for the unique properties of both components.

## 3 METHOD

**Notations** We first describe the forecasting task with the sliding window approach (Zhou et al., 2021; Wu et al., 2021; Park et al., 2023), which covers all possible in-output sequence pairs of the entire time series $\mathcal{S} = \{\mathbf{s}_1, \ldots, \mathbf{s}_T\}$ where $T$ denotes the length of the observed time series and $\mathbf{s}_t \in \mathbb{R}^c$ is observation with $c$ dimension. For the simplicity in explaining our methodology, we set the dimension $c$ to 1 throughout this paper. By sliding a window with a fixed length $W$ on $\mathcal{S}$, we obtain the windows $\mathcal{D} = \{\mathcal{W}_t\}_{t=1}^M$ where $\mathcal{W}_t = (\mathcal{X}_t, \mathcal{Y}_t)$ are divided into two parts: input sequence $\mathcal{X}_t = \{\mathbf{s}_t, \ldots, \mathbf{s}_{t+I-1}\}$ with the input length $I$, and output sequence $\mathcal{Y}_t = \{\mathbf{s}_{t+I}, \ldots, \mathbf{s}_{t+I+O-1}\}$ with the output length $O$ to predict. Also, we denote the global index sequence of $\mathcal{W}_t$ as $\mathcal{T}_t = \{t + i\}_{i=0}^{W-1}$.

### 3.1 AUTOCORRELATION-BASED CONTRASTIVE LOSS FOR LONG-TERM FORECASTING

**Missing Long-term Dependency in the Window** Many real-world time series exhibit diverse long-term and short-term variations (Wu et al., 2021; 2023; Wang et al., 2023). In such cases, a forecasting model may struggle to predict long-term variations, as these variations are not captured within the window. We first identify these long-term variations using autocorrelation, inspired by the stochastic process theory(Chatfield & Xing, 2019; Papoulis & Unnikrishna Pillai, 2002). For a real discrete-time process $\{\mathcal{S}_t\}$, we can obtain the autocorrelation function $\mathcal{R}_{\mathcal{SS}}(h)$ using the following equation:

$$\mathcal{R}_{\mathcal{SS}}(h) = \lim_{T \to \infty} \frac{1}{T} \sum_{t=1}^{T} \mathcal{S}_t \mathcal{S}_{t-h} \tag{1}$$

The autocorrelation measures the correlation between observations at different times (*i.e.,* time lag $h$). A strong correlation close to 1 or -1 indicates that all points separated by $h$ in the series $\mathcal{S}$ are linearly related, moving in the same or opposite direction for positive or negative signs, respectively. In other words, autocorrelation can be utilized to forecast future variations that are $h$ interval away based on current variations. Although recent methods have leveraged autocorrelation to discover period-

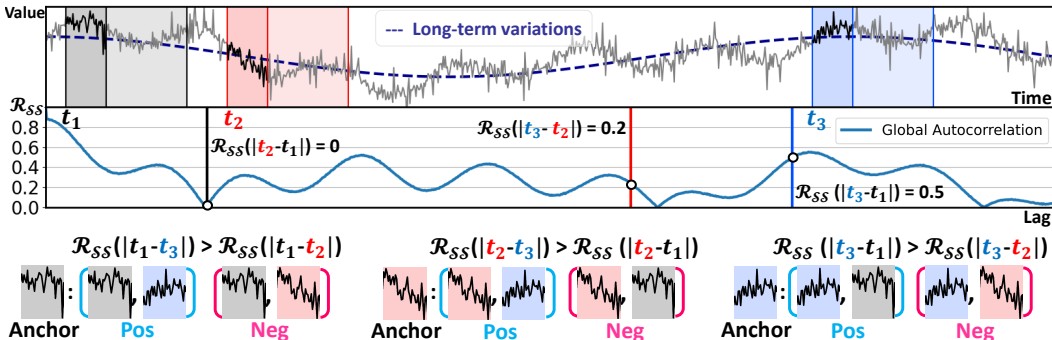

Figure 3: Example of the relative selection strategy in our AutoCon. Three windows are sampled from different times $t_1$, $t_2$, and $t_3$ on the entire series to make up the batch. In this batch, there are a total of three possible positive pairs (*i.e.,* due to three anchors). Each pair calculates a global autocorrelation whose lag is the time distance of the two windows constituting the pair. Then, by comparing the autocorrelation with other remaining pairs, the pairs with lower autocorrelation than the anchor positive pair are designated as negative pairs.

based dependencies (Wu et al., 2021; Wang et al., 2022), they only apply it to capture variations within the window, overlooking long-term variations that span beyond the window. But as shown in Figure 1, non-zero correlations exist outside the conventional window length. For the first time, we propose a representation learning method via contrastive learning to capture these long-term variations quantified by the global autocorrelation. Note that, to distinguish our method from those that use local autocorrelation within a given window, we refer to the autocorrelation calculated across the entire time series as the global autocorrelation.

**Autocorrelation-based Contrastive Loss (AutoCon)**   We note that a mini-batch can consist of windows that are temporally very far apart. This time distance can be as long as the entire series length $T$, which is much longer than the window length $W$. Based on this fact, we address long-term dependencies that exist throughout the entire series by establishing relationships between windows. Concretely, we define the relationship between the two windows based on the global autocorrelation. Any two windows $\mathcal{W}_{t_1}$ and $\mathcal{W}_{t_2}$ obtained at two different times $t_1$ and $t_2$ each have $W$ observations with globally indexed time sequence $\mathcal{T}_{t_1} = \{t_1 + i\}_{i=0}^{W-1}$ and $\mathcal{T}_{t_2} = \{t_2 + j\}_{j=0}^{W-1}$. Then, we denote time distances between all pairs of two observations in each window as a matrix $\boldsymbol{D} \in \mathbb{R}^{W \times W}$. This matrix $\boldsymbol{D}$ contains time distance as elements $\boldsymbol{D}_{i,j} = |(t_2 + j) - (t_1 + i)|$. In the two windows, the time distances between the same phase (*i.e.,* $i = j$) all have the same value $|t_1 - t_2|$, and they are represented by the diagonal terms $\{\boldsymbol{D}_{i,i}\}_{i=1}^{W-1}$ of the matrix. Therefore, based on this representativeness, we leverage the global autocorrelation $\mathcal{R}_{SS}(|t_1 - t_2|)$ to define the relationship between the two windows as follows:

$$r(\mathcal{T}_{t_1}, \mathcal{T}_{t_2}) = |\mathcal{R}_{SS}(|t_1 - t_2|)| \tag{2}$$

where $\mathcal{R}_{SS}$ denote the global autocorrelation calculated from train series $\mathcal{S}$.

Now, we design a loss to ensure that the similarities between all pairs of window representations follow the global autocorrelation measured in the data space. To achieve this, we define positive and negative samples in a relative manner inspired by SupCR (Zha et al., 2022) for regression tasks in the image domain. However, unlike SupCR which uses annotated labels to determine the relationship between images, we use the global autocorrelation $\mathcal{R}_{SS}$ to determine the relationship between windows, making our approach an unsupervised method. We feed a mini-batch $\mathcal{X} \in \mathbb{R}^{N \times I}$ consisting of $N$ windows to the encoder to obtain representations $\boldsymbol{v} \in \mathbb{R}^{N \times I \times d}$ where $\boldsymbol{v} = Enc\,(\mathcal{X}, \mathcal{T})$. Indexed by the windows $i$, our autocorrelation-based contrastive loss, called AutoCon, is then computed over the representations $\{\boldsymbol{v}^{(i)}\}_{i=1}^{N}$ with the corresponding time sequence $\{\mathcal{T}^{(i)}\}_{i=1}^{N}$ as:

$$\mathcal{L}_{\text{AutoCon}} = -\frac{1}{N} \sum_{i=1}^{N} \frac{1}{N-1} \sum_{j=1, j \neq i}^{N} r^{(i,j)} \log \frac{\exp\left(Sim\left(\boldsymbol{v}^{(i)}, \boldsymbol{v}^{(j)}\right)/\tau\right)}{\sum_{k=1}^{N} \mathbb{1}_{\left[k \neq i, r^{(i,k)} \leq r^{(i,j)}\right]} \exp\left(Sim\left(\boldsymbol{v}^{(i)}, \boldsymbol{v}^{(k)}\right)/\tau\right)} \tag{3}$$

where $Sim\,(\cdot, \cdot)$ measures the similarity between two representations (*e.g.,* cosine similarity between max pooled $\boldsymbol{v}^{(i)}$ along with the time axis (Yue et al., 2022)), and $r^{(i,j)} = r(\mathcal{T}^{(i)}, \mathcal{T}^{(j)})$ represents

the global correlation between two windows. During training, there are a total of $N \times (N-1)$ possible pairs indexed by $(i, j)$. Each pair (*i.e.,* as an anchor pair) designates itself as a relatively positive pair by considering any pairs that exhibit the global autocorrelation $r^{(i,k)}$ lower than that $r^{(i,j)}$ of the anchor pair as negative pairs. Figure 3 describes sample cases of our selection strategy in the given batch. Since a different set of windows form the batch in each iteration, we expect that the representations reflect the global autocorrelations of all possible distances. The relative selection strategy does not guarantee that the positive window has a high correlation close to one; it only requires a higher correlation than other negative windows in the same batch. Consequently, we introduce $r^{(i,j)}$ as weights to differentiate between positive pairs with varying degrees of correlation, similar to focal loss (Lin et al., 2017). To minimize $\mathcal{L}_{\text{AutoCon}}$, the encoder learns representations so that the pairs with high correlation are closer than the pairs with low correlation.

Our AutoCon offers several notable advantages over conventional contrastive-based methods. First, although AutoCon is an unsupervised representation method, it does not rely on data augmentation, which is common in most contrastive-based approaches (Tonekaboni et al., 2021; Yue et al., 2022; Woo et al., 2022a). The augmentation-based methods require additional computation costs caused by the augmentation process and increase the forward-backward process for the augmented data. Also, existing contrastive learning methods consider only temporally close samples as positive pairs within windows. This ultimately fails to appropriately learn representations of the windows that are distant from each other but are similar due to long-term periodicity. Consequently, our method is computationally efficient and able to learn long-term representations, enhancing the ability to predict long-term variations effectively.

## 3.2 Decomposition Architecture for Long-term Representation

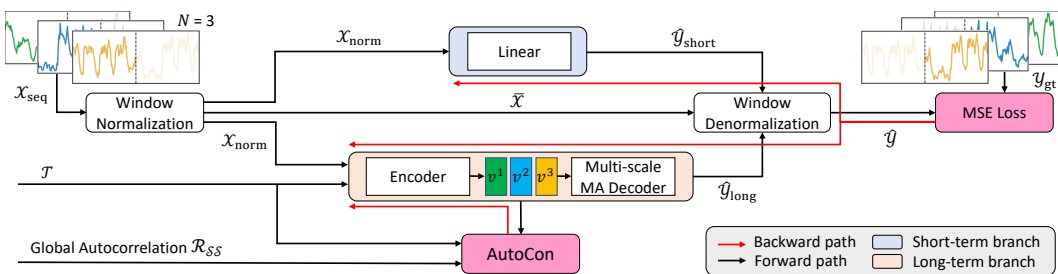

Figure 4: An overview of the redesigned architecture for long-term representation and forecasting

Existing models commonly adopt the decomposition architecture that has a seasonal branch and a trend branch to achieve disentangled seasonal and trend prediction. To emphasize that trends are long-term variations partially caught in the window, we regard the trend branch as a long-term branch and the seasonal branch as a short-term branch. Our AutoCon method is designed to learn long-term representations, making it natural not to use it in the short-term branch to enforce long-term dependencies. However, integrating AutoCon with current decomposition architectures presents a challenge because both branches share the same representation (Wu et al., 2021; Zhou et al., 2022b; Liu et al., 2022b), or the long-term branch consists of a linear layer that is not suitable for learning representations (Zeng et al., 2023; Wang et al., 2023). Moreover, we observe that recent linear-based models (Zeng et al., 2023) outperform complicated deep models at short-term predictions, leaving doubts whether a deep model is necessary to learn the high-frequency variations. Based on these considerations, we redesign a model architecture with well-defined existing blocks to respect temporal locality for short-term and globality for long-term forecasting as shown in Figure 4. Our decomposition architecture has three main features.

**Normalization and Denormalization for Nonstationarity** First, we use window-unit normalization and denormalization methods (Equation 4) (Kim et al., 2021; Zeng et al., 2023) as follows:

$$\mathcal{X}_{norm} = \mathcal{X} - \bar{\mathcal{X}}, \qquad \mathcal{Y}_{pred} = (\mathcal{Y}_{short} + \mathcal{Y}_{long}) + \bar{\mathcal{X}} \tag{4}$$

where $\bar{\mathcal{X}}$ is the mean of the input sequence. These simple methods help to effectively alleviate the distribution shift problem (Kim et al., 2021) by nonstationarity of the real-world time series.

**Short-term Branch for Temporal Locality** Next, we observe that short-period variations often repeat multiple times within the input sequence and exhibit similar patterns with temporally close

sequences. This locality of short-term variations supports the recent success of linear-based models (Zeng et al., 2023), which use sequential information of adjacent sequences only. Therefore, we employ the linear layer for the short-term prediction as follows:

$$\mathcal{Y}_{short} = Linear(\mathcal{X}_{norm}). \tag{5}$$

**Long-term Branch for Temporal Globality**   The long-term branch, designed to apply the AutoCon method, employs an encoder-decoder architecture. The encoder with sufficient capacity to learn the long-term presentation leverages both sequential information and global information (*i.e.,* timestamp-based features derived from $\mathcal{T}$) as follows:

$$\boldsymbol{v} = Enc(\mathcal{X}_{norm}, \mathcal{T}). \tag{6}$$

The choice of network for the encoder is flexible as long as there are no issues in processing long sequences. We opted for temporal convolution networks (Bai et al., 2018) (TCNs), widely used in learning time-series representation (Yue et al., 2022), for its computational efficiency. The decoder employs the multi-scale Moving Average (MA) block (Wang et al., 2023), with different kernel sizes $\{k_i\}_{i=1}^{n}$ to capture multiple periods based on the representation $\boldsymbol{v}$ as follows:

$$\hat{\mathcal{Y}}_{long} = \frac{1}{n} \sum_{i=1}^{n} AvgPool(Padding(MLP(\boldsymbol{v})))_{k_i}. \tag{7}$$

The MA block at the head of the long-term branch smooths out short-term fluctuations, naturally encouraging the branch to focus on long-term information. Our redesigned architecture is optimized by the objective function $\mathcal{L}$ as follows:

$$\mathcal{L} = \mathcal{L}_{\text{MSE}} + \lambda \cdot \mathcal{L}_{\text{AutoCon}}. \tag{8}$$

where the mean square error (MSE) and the AutoCon loss are combined with the weight $\lambda$ as a hyperparameter. The hyperparamer sensitivity analysis is available in Appendix A.6. Detailed descriptions of each operation (*e.g., $Linear$, $Padding$,* and $MLP$) can be found in the Appendix A.1.

## 4   EXPERIMENTS

To validate our proposed method, we conducted extensive experiments on nine real-world datasets from six domains: mechanical systems (ETT), energy (Electricity), traffic (Traffic), weather (Weather), economics (Exchange), and disease (ILI). We follow standard protocol (Wu et al., 2021) and split all datasets into training, validation, and test sets in chronological order by the ratio of 6:2:2. We select the latest baseline models with different architectures categorized into linear-based (Zhou et al., 2022a; Zeng et al., 2023), CNN-based (Wu et al., 2023; Wang et al., 2023), and Transformer-based (Zhou et al., 2022b; Liu et al., 2022b; Nie et al., 2023). Additionally, we compared our model with two models (Challu et al., 2023; Zhang & Yan, 2023) that focus on learning inter-channel dependencies for multivariate forecasting. Appendix A provides more detailed information about the datasets and baseline implementations.

### 4.1   MAIN RESULTS

**Extended Long-term Forecasting**   In our pursuit to better evaluate our model's performance in predicting long-term variations—which tend to have increasing significance as the forecast length extends—we designed our experiments to extend the prediction length $O$ for each dataset. This shift from the conventional benchmark experiments, which typically predict up to 720 steps, allows us to explore the model's capability in more challenging forecast scenarios. For the datasets with longer total lengths, such as ETTh, Electricity, Traffic, and Weather, we tripled the prediction length from 720 to 2160. Also, for Exchange and ILI datasets with shorter total lengths, we extend the output length up to 1080 and 112, respectively. Overall, Table 1 shows that our model with AutoCon outperformed the state-of-the-art baselines by achieving first place 42 times in the univariate setting. When examining the performance changes according to length, our model showed significant improvement compared to other best models when predicting further into the future (*e.g.,* on average, errors decreased by 5% at 96 and 720, and by 12% at 1440 and 2160). These results empirically demonstrate the contribution of our AutoCon in effectively capturing long-term variations that exist beyond the window.

**Dataset Analysis**   Since our goal is to learn long-term variations, the performance improvements of our model can be affected by the magnitude and the number of long-term variations. Figure 5 shows various yearly-long business cycles and natural cycles unique to each dataset. For instance,

Table 1: Extended long-term forecasting results with various prediction lengths $O$ and the best input length $I \in \{48, 96, 168, 336\}$ for each model except for Illness dataset with $I = 14$. Red and blue numbers denote the best and second-best results, respectively. The full benchmarks with ETTh1 and ETTm are available in Appendix D.

| Models | | Ours | | TimesNet (2023) | | MICN (2023) | | PatchTST (2023) | | DLinear (2023) | | FiLM (2022a) | | Nonstationary (2022b) | | FEDformer (2022b) | |
|---|---|---|---|---|---|---|---|---|---|---|---|---|---|---|---|---|---|
| | $O$ | MSE | MAE | MSE | MAE | MSE | MAE | MSE | MAE | MSE | MAE | MSE | MAE | MSE | MAE | MSE | MAE |
| ETTh2 | 96 | 0.124 | 0.269 | 0.139 | 0.290 | 0.122 | 0.264 | 0.136 | 0.292 | 0.128 | 0.271 | 0.129 | 0.275 | 0.192 | 0.343 | 0.129 | 0.277 |
| | 720 | 0.177 | 0.344 | 0.207 | 0.370 | 0.313 | 0.457 | 0.233 | 0.392 | 0.319 | 0.461 | 0.256 | 0.407 | 0.231 | 0.394 | 0.273 | 0.419 |
| | 1440 | 0.176 | 0.340 | 0.192 | 0.358 | 0.520 | 0.599 | 0.351 | 0.481 | 0.514 | 0.597 | 0.389 | 0.506 | 0.211 | 0.379 | 0.384 | 0.487 |
| | 2160 | 0.198 | 0.358 | 0.263 | 0.413 | 0.759 | 0.734 | 0.610 | 0.659 | 0.740 | 0.728 | 0.610 | 0.645 | 0.240 | 0.399 | 0.919 | 0.737 |
| Electricity | 96 | 0.196 | 0.313 | 0.286 | 0.386 | 0.241 | 0.367 | 0.227 | 0.336 | 0.207 | 0.322 | 0.394 | 0.451 | 0.332 | 0.426 | 0.279 | 0.393 |
| | 720 | 0.275 | 0.386 | 0.417 | 0.471 | 0.336 | 0.446 | 0.332 | 0.426 | 0.304 | 0.412 | 0.467 | 0.504 | 0.505 | 0.533 | 0.417 | 0.486 |
| | 1440 | 0.338 | 0.441 | 0.491 | 0.523 | 0.419 | 0.504 | 0.482 | 0.537 | 0.395 | 0.484 | 0.625 | 0.610 | 0.577 | 0.574 | 0.651 | 0.609 |
| | 2160 | 0.380 | 0.481 | 0.536 | 0.547 | 0.421 | 0.501 | 0.768 | 0.644 | 0.415 | 0.496 | 0.938 | 0.758 | 0.642 | 0.610 | 0.896 | 0.714 |
| Traffic | 96 | 0.132 | 0.206 | 0.145 | 0.219 | 0.168 | 0.256 | 0.192 | 0.296 | 0.219 | 0.327 | 0.264 | 0.334 | 0.247 | 0.326 | 0.220 | 0.312 |
| | 720 | 0.144 | 0.225 | 0.163 | 0.269 | 0.304 | 0.394 | 0.213 | 0.318 | 0.309 | 0.419 | 0.247 | 0.329 | 0.277 | 0.360 | 0.255 | 0.344 |
| | 1440 | 0.174 | 0.251 | 0.188 | 0.292 | 0.375 | 0.443 | 0.246 | 0.341 | 0.353 | 0.409 | 0.311 | 0.390 | 0.303 | 0.361 | 0.297 | 0.376 |
| | 2160 | 0.175 | 0.252 | 0.190 | 0.304 | 0.360 | 0.426 | 0.261 | 0.353 | 0.324 | 0.386 | 0.988 | 0.745 | 0.222 | 0.317 | 0.317 | 0.394 |
| Weather | 96 | 0.521 | 0.522 | 0.584 | 0.536 | 0.569 | 0.525 | 0.545 | 0.539 | 0.579 | 0.529 | 0.589 | 0.533 | 0.636 | 0.567 | 0.703 | 0.625 |
| | 720 | 0.963 | 0.715 | 1.090 | 0.753 | 1.080 | 0.754 | 0.987 | 0.752 | 1.007 | 0.706 | 1.003 | 0.728 | 1.007 | 0.725 | 1.114 | 0.822 |
| | 1440 | 1.280 | 0.835 | 1.547 | 0.926 | 1.351 | 0.863 | 1.342 | 0.860 | 1.299 | 0.823 | 1.472 | 0.900 | 1.394 | 0.867 | 1.435 | 0.919 |
| | 2160 | 1.415 | 0.887 | 1.744 | 0.994 | 1.544 | 0.937 | 1.506 | 0.924 | 1.454 | 0.887 | 1.712 | 0.988 | 1.598 | 0.944 | 1.786 | 1.054 |
| Exchange | 48 | 0.051 | 0.172 | 0.054 | 0.178 | 0.054 | 0.181 | 0.068 | 0.197 | 0.049 | 0.170 | 0.052 | 0.173 | 0.054 | 0.178 | 0.059 | 0.184 |
| | 360 | 0.448 | 0.527 | 0.479 | 0.532 | 0.459 | 0.536 | 0.548 | 0.573 | 0.485 | 0.531 | 0.492 | 0.534 | 0.493 | 0.541 | 0.528 | 0.556 |
| | 720 | 1.067 | 0.794 | 1.239 | 0.856 | 1.383 | 0.927 | 1.264 | 0.859 | 1.718 | 1.024 | 1.291 | 0.864 | 1.358 | 0.894 | 1.381 | 0.903 |
| | 1080 | 1.004 | 0.792 | 1.327 | 0.900 | 4.874 | 1.972 | 1.255 | 0.873 | 4.982 | 1.973 | 1.670 | 1.010 | 1.774 | 1.058 | 1.600 | 0.980 |
| ILI | 14 | 0.725 | 0.574 | 1.414 | 0.735 | 0.815 | 0.701 | 1.558 | 0.965 | 1.397 | 0.901 | 1.079 | 0.739 | 1.107 | 0.698 | 0.773 | 0.619 |
| | 28 | 0.887 | 0.683 | 1.604 | 0.854 | 1.670 | 1.062 | 1.878 | 1.110 | 2.008 | 1.134 | 1.315 | 0.887 | 1.515 | 0.767 | 0.989 | 0.770 |
| | 56 | 0.807 | 0.725 | 1.021 | 0.787 | 1.757 | 1.210 | 1.451 | 1.028 | 1.584 | 1.075 | 1.080 | 0.891 | 0.895 | 0.742 | 0.856 | 0.741 |
| | 112 | 1.499 | 1.038 | 1.669 | 1.072 | 3.593 | 1.759 | 2.846 | 1.438 | 3.332 | 1.572 | 2.608 | 1.387 | 1.724 | 1.108 | 1.660 | 1.097 |
| $1^{st}$ Count | | 42 | | 0 | | 2 | | 0 | | 4 | | 0 | | 0 | | 0 | |

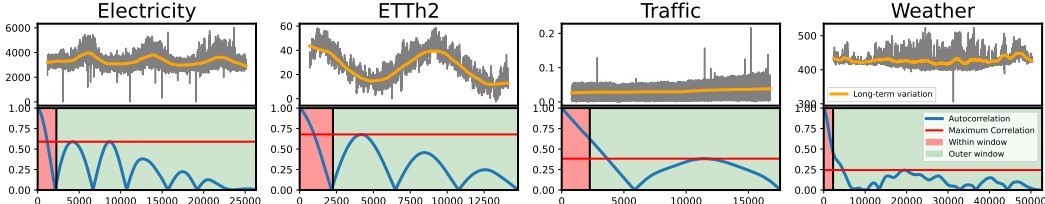

Figure 5: The outer-window autocorrelation exists in varying degrees in four datasets.

the ETTh2 and Electricity datasets have strong long-term correlations with peaks at several lags repeated multiple times. Thus, our method on the ETTh2 and Electricity datasets exhibited significant performance gains, which are 34% and 11% reduced error compared to the second-best model, respectively. In contrast, the Weather dataset has relatively lower correlations outside the windows than the aforementioned two datasets. This leads our model to show the least improvement with a 3% reduced error on the Weather dataset. As a result, our method's superiority manifests more strongly for the datasets with stronger long-term correlation, thus empirically validating our contribution.

**Extension to Multivariate Forecasting** As shown in Table 2, our method is applicable in multivariate forecasting by calculating autocorrelation on a per-channel basis and then following a channel independence approach (Nie et al., 2023). Appendix A.2 describes details for multivariate setting.

### 4.2 MODEL ANALYSIS

**Temporal Locality and Globality** As mentioned in Section 3.2, we proposed a model architecture that combines the advantages of both linear models for locality and deep models for globality. Figure 6(a) demonstrates that, for short-term predictions up to 96 units, the linear model (DLinear) achieved a lower error rate than the deep models such as TimesNet, Nonstationary, and FEDformer. However, the error of DLinear started to diverge as the prediction length extended. Conversely, the TimesNet and Nonstationary maintained a consistent error rate even with the increase in prediction length but didn't perform as well as the linear model for short-term predictions. These observations served as the motivation for our decomposition architecture that is proficient in both short-term and long-term predictions (blue line in Figure 6(a)).

**Ablation Studies** Here, we conducted ablation studies to validate each component of our method. Figure 6(b) shows the results of the ablation study conducted on the full model, and when the

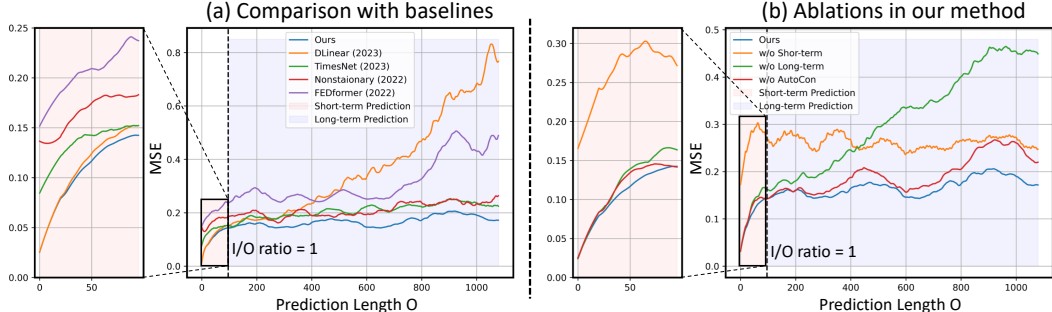

Figure 6: Comparison of forecasting error (MSE) according to prediction length $O$. (a) comparison with baseline models and (b) comparison between ablations of our method on the ETTh2 dataset.

Table 2: Multivariate forecasting results on ETT datasets with different prediction lengths $O \in \{96, 192, 336, 720\}$ and the input length $I = 96$. Due to lack of space, we report the averaged performance of four length settings. The full benchmark is available at Appendix D.

| Models | | Ours | TimesNet* (2023) | | MICN (2023) | | Crossformer (2023) | | N-HiTS (2023) | | DLinear* (2023) | | ETSformer* (2022b) | | LightTS* (2022) | |
|---|---|---|---|---|---|---|---|---|---|---|---|---|---|---|---|---|---|
| Dataset | MSE | MAE | MSE | MAE | MSE | MAE | MSE | MAE | MSE | MAE | MSE | MAE | MSE | MAE | MSE | MAE |
| ETTh1 | **0.442** | **0.431** | 0.458 | 0.450 | 0.559 | 0.535 | 0.591 | 0.550 | 0.502 | 0.490 | 0.456 | 0.452 | 0.542 | 0.510 | 0.491 | 0.479 |
| ETTh2 | **0.372** | **0.401** | 0.414 | 0.427 | 0.588 | 0.525 | 0.885 | 0.673 | 0.545 | 0.491 | 0.559 | 0.515 | 0.439 | 0.452 | 0.602 | 0.543 |
| ETTm1 | **0.390** | **0.400** | 0.400 | 0.406 | 0.392 | 0.414 | 0.503 | 0.489 | 0.428 | 0.436 | 0.403 | 0.407 | 0.429 | 0.425 | 0.435 | 0.437 |
| ETTm2 | **0.281** | **0.325** | 0.291 | 0.333 | 0.328 | 0.382 | 0.593 | 0.535 | 0.346 | 0.383 | 0.350 | 0.401 | 0.293 | 0.342 | 0.409 | 0.436 |

∗ denotes the results, which are taken from TimesNet (Wu et al., 2023).

short-term branch was removed, it showed a significant error for short-term predictions. When the long-term branch was removed, it showed a significant error for long-term predictions. Also, without the integration of our AutoCon, the long-term performance was degraded. As demonstrated in Table 3, these trends were consistent across a variety of datasets.

## 4.3 COMPARISON WITH REPRESENTATION LEARNING METHODS

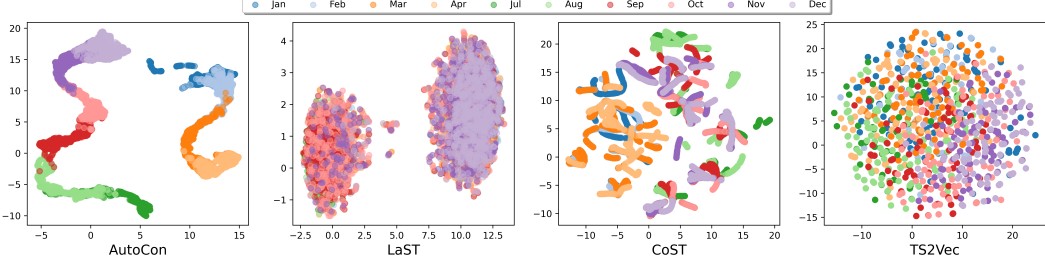

Figure 7: The figure displays UMAP (McInnes et al., 2018) visualization results over the representations of different four methods on the ETTh2 dataset. Our AutoCon demonstrates clear continuity and clustering between adjacent months, indicating an understanding of long-term variation. In contrast, the other models appear to lack this perceptible one-year long-term structure, possibly due to the limited representation learning within the window.

We also demonstrate the effectiveness of our method in capturing long-term representations beyond the window compared to existing time-series representation learning methods. TS2Vec (Yue et al., 2022) and CoST (Woo et al., 2022a) are both unsupervised contrastive learning methods, with TS2Vec only considering the augmented data of the same time index as a positive pair and CoST using a loss that takes into account periodicity, but both have the limitation of only being effective within

Table 3: Ablation of the short-term, long-term branch, and AutoCon in Ours.

| Datasets | | ETTh1 | | | | ETTh2 | | | | ETTm2 | | | |
|---|---|---|---|---|---|---|---|---|---|---|---|---|---|
| Prediction length | | 96 | 720 | 1440 | 2160 | 96 | 720 | 1440 | 2160 | 192 | 1440 | 2880 | 4320 |
| Ours | MSE | **0.055** | **0.078** | **0.078** | **0.074** | **0.125** | **0.177** | **0.176** | **0.198** | **0.093** | **0.214** | **0.211** | **0.214** |
| w/o Short-term | MSE | 0.071 | 0.093 | 0.126 | 0.094 | 0.204 | 0.271 | 0.263 | 0.302 | 0.186 | 0.313 | 0.300 | 0.288 |
| w/o Long-term | MSE | 0.055 | 0.084 | 0.093 | 0.108 | 0.126 | 0.242 | 0.353 | 0.592 | 0.093 | 0.235 | 0.258 | 0.326 |
| w/o AutoCon | MSE | 0.061 | 0.082 | 0.096 | 0.130 | 0.147 | 0.214 | 0.212 | 0.236 | 0.118 | 0.302 | 0.254 | 0.237 |

Table 4: Comparison with representation learning methods.

| Datasets | | ETTm1 | | | | Exchange | | | | ILI | | | |
|---|---|---|---|---|---|---|---|---|---|---|---|---|---|
| Prediction length | | 192 | 1440 | 2880 | 4320 | 48 | 360 | 720 | 1080 | 14 | 28 | 56 | 112 |
| AutoCon | MSE | **0.041** | **0.090** | **0.089** | **0.082** | **0.051** | 0.448 | **1.067** | **1.003** | **0.724** | **0.886** | **0.810** | **1.499** |
| LaST (2022) | MSE | 0.053 | 0.120 | 0.204 | 0.274 | 0.051 | **0.418** | 2.022 | 5.529 | 1.730 | 2.712 | 1.694 | 3.206 |
| CoST (2022) | MSE | 0.059 | 0.130 | 0.199 | 0.192 | 0.054 | 0.451 | 1.703 | 4.470 | 0.746 | 1.114 | 1.490 | 3.155 |
| TS2Vec (2022) | MSE | 0.064 | 0.150 | 0.171 | 0.161 | 0.059 | 0.516 | 1.387 | 5.337 | 2.161 | 2.262 | 3.043 | 4.098 |

a window. Therefore, while they show competitive performance in relatively short lengths, they fail to predict accurately for long-term periods. LaST (Wang et al., 2022) is a decomposition-based representation learning method and also shows competitive performance in short-term predictions, but fails to predict accurately for long-term periods. Figure 7 shows the learned representation by AutoCon with the other three methods. Appendix C.2 provides experimental protocols and further comparison experiments.

### 4.4 COMPUTATIONAL EFFICIENCY COMPARISON

Our proposed model shows competitive computational efficiency among other deep models. Specifically, on the ETT dataset, our model without AutoCon exhibits computational times of 31.1 ms/iter, second best after the linear models. Even with the integration of AutoCon during training, the computational cost does not increase significantly (33.2 ms/iter) since there is no augmentation process and the autocorrelation calculation occurs only once during the entire training. Consequently, our model's computational efficiency surpasses existing Transformer-based models (Nonstationary 365.7 ms/iter) and recent state-of-the-art CNN-based models (TimesNet 466.1 ms/iter). Detailed comparisons can be found in Appendix B.4.

### 5 DISCUSSION & LIMITATIONS

Our proposed method mitigates the constraint of the sliding window approach by learning the long-term variations beyond the window. Nevertheless, we examine whether the limitation of the sliding window we point out can be solved by simply increasing the window length without our method, and also elucidate a limitation of our method.

**Can we use a very long window to capture long-term variations?** Given a time series $\mathcal{S}$ of length $T$, the number of windows $M$ is $T - (I + O) + 1$. This implies that as the input length $I$ (*i.e.,* data complexity) increases, while keeping the output length $O$ fixed, the number of data instances (*i.e.,* windows) available for learning decreases, potentially making the model more susceptible to overfitting (Park et al., 2023) as shown in Figure 8. Consequently, it is challenging for input

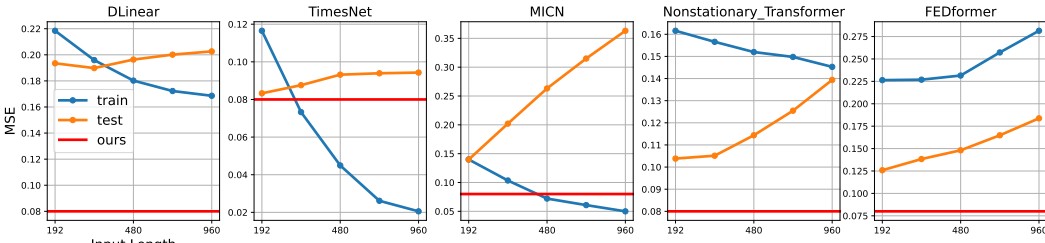

Figure 8: The figure shows the training (blue line) and test (orange line) MSE loss trace plots of five models as the input length gradually increases from 192 to 960 on the ETTh1 with the output-720 setting. In the figure, the red horizontal line represents our test loss with the input-96 setting.

sequences to be long enough to cover all long-term variations present in the data, and models often struggle to capture variations outside the window. Therefore, the limitation we point out regarding the sliding window approach is valid in most situations and worth addressing. Appendix B.1 presents comprehensive experiments and empirical findings obtained upon increasing the window length.

**Can Autocorrelation capture all long-term variations ?** While autocorrelation serves as a valuable tool for capturing certain long-term variations, its linearity assumption limits its effectiveness in dealing with the non-linear patterns and relationships prevalent in real-world time series data. By considering higher-order correlations, non-linear dependencies, and external factors, we are likely to achieve even more accurate and comprehensive long-term forecasting.

ACKNOWLEDGMENTS

This work was supported by the National Research Foundation of Korea (NRF) grant funded by the Korea government (MSIT) (No.NRF-2020H1D3A2A03100945, No.NRF-2022R1A2B5B02001913), and Institute of Information & communications Technology Planning & Evaluation (IITP) grant funded by the Korea government (MSIT) (No.2019-0-00075, No.2022-0-00984).

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

## A  REPRODUCIBILITY

### A.1  DETAILS OF OUR METHOD IMPLEMENTATION

In this subsection, we provide a detailed explanation of the operations that were used in Section 3.2. Firstly, $Linear$ signifies a linear layer along the time dimension. When provided with an input sequence $\mathcal{X}$, the output $\hat{\mathcal{Y}}$ is computed as:

$$\hat{\mathcal{Y}} = W_{time} \cdot \mathcal{X} \tag{9}$$

where $\mathcal{X} \in \mathbb{R}^{I \times c}$, $\hat{\mathcal{Y}} \in \mathbb{R}^{O \times c}$, and $W_{time} \in \mathbb{R}^{O \times I}$.

Next, we describe $Padding(\cdot)$ and $Avgpool(\cdot)$ along with the time axis as follows:

$$\mathcal{X}_{pad} = Padding(\mathcal{X}) \tag{10}$$

$$\mathcal{X}_{avg} = Avgpool(\mathcal{X}_{pad})_k \tag{11}$$

$$\mathcal{X}_{avg}[t, c] = \frac{1}{k} \sum_{i=0}^{k-1} \mathcal{X}_{pad}[t + i, c] \tag{12}$$

where $\mathcal{X}_{pad} \in \mathbb{R}^{(I+2(k-1)) \times c}$ is padded sequence on both sides with neighboring values to preserve input length after applying $Avgpool(\cdot)$. The various kernel sizes of $Avgpool(\cdot)$ are selected to handle the multi-periodicities, which are observed in real-world time-series data.

$MLP$ denotes the use of two linear layers with an activation function $g$, specifically GELU. Given the input sequence $\mathcal{X}$, the output $\mathcal{Y}$ is computed as:

$$\hat{\mathcal{Y}} = g(W_{time} \cdot \mathcal{X} + b_{time}) \cdot W_{channel} + b_{channel} \tag{13}$$

where $b_{time} \in \mathbb{R}^d$, $b_{channel} \in \mathbb{R}^c$, $W_{time} \in \mathbb{R}^{O \times I}$, and $W_{channel} \in \mathbb{R}^{d \times c}$.

To compute AutoCon, global autocorrelation must first be determined. As part of the preprocessing process, we calculate autocorrelation for the entire training series, excluding validation and test series. This autocorrelation comprises both long and short-period variations. However, in our pursuit of disentangled long-term representation, we intend that the long-term branch address only low-frequency variations with long periods. Therefore, we smooth out short-period fluctuations in the series before computing the autocorrelation.

Our training protocol is identical to conventional training methods, except for the inclusion of AutoCon as an additional loss, in addition to the forecasting loss. To clarify this further, we also present the algorithm.

---

**Algorithm 1 AutoCon**: **Auto**correlation-based **Con**strastive Learning Framework

---

**Require:** Entire time series $\mathcal{S} = \{s_1, \ldots, s_T\}$, Training set $\mathcal{D} = \{(\mathcal{X}_t, \mathcal{Y}_t), \mathcal{T}_t\}_{t=1}^{M}$,
   AutoCon weight $\lambda$ where $T$ denotes a total length and $M$ denotes the number of windows.

   Compute the global autocorrelation $\mathcal{R}_{\mathcal{SS}}(l)$ for all possible lags $l \in [0 : M]$

   **for all** number of training iterations **do**
      Sample a mini-batch $\{((\mathcal{X}_n, \mathcal{Y}_n), \mathcal{T}_n)\}_{n=1}^{N}$ from $\mathcal{D}$
      Forward $\{\mathcal{X}_n, \mathcal{T}_n\}_{n=1}^{N}$ and get corresponding representation $\{v\}_{n=1}^{N}$ and predictions $\{\hat{\mathcal{Y}}_n\}_{n=1}^{N}$
      Compute window relationship matrix $r \in \mathbb{R}^{N \times N}$, $r^{(i,j)} = \mathcal{R}_{\mathcal{SS}}(|\mathcal{T}_i^{(0)} - \mathcal{T}_j^{(0)}|)$
      Compute AutoCon loss $\mathcal{L}_{\text{AutoCon}}$ following Equation 3 as inputs $\{v^{(n)}\}_{n=1}^{N}$ and $r$
      Compute forecasting loss $\mathcal{L}_{mse} = \frac{1}{N} \sum_{n=1}^{N} (\hat{\mathcal{Y}}_n - \mathcal{Y}_n)^2$
      Do one training step using the full loss $\mathcal{L} = \mathcal{L}_{mse} + \lambda \cdot \mathcal{L}_{\text{AutoCon}}$
   **end for**

---

Our redesigned model and AutoCon were implemented based on the TSlib code repository [1]. Our source code can be accessed at a zip file in the supplementary.

## A.2 DETAILS OF MULTIVARIATE FORECASTING

There are two representative approaches to multivariate forecasting: *Channel-mixing* and *Channel-independence* approaches. The channel-mixing approach involves mapping the values of multiple channels at the same step into an embedding space and extracting temporal dependencies from this embedding sequence. This approach has been adopted by various papers (Zhou et al., 2022b; Wu et al., 2023; Zhang & Yan, 2023). The channel-independence approach, on the other hand, preserves each channel's information without mixing them and learns temporal patterns within each channel independently. Recently, this method has been adopted in high-performing models such as PatchTST (Challu et al., 2023) and Linear models (Zeng et al., 2023), demonstrating superior performance on current benchmark datasets. In terms of implementation, each channel is treated as a batch axis for computations. This effectively increases the amount of training data by the number of channels, and the model parameters are shared across multiple channels. Following the channel- independence approach, we first compute autocorrelations for each channel separately in order to calculate AutoCon. We then train the representation tailored to each channel based on these autocorrelations.

## A.3 DETAILS OF DATASETS

In this paper, we utilized six real-world datasets from diverse domains: mechanical systems (ETT), energy (Electricity), traffic (Traffic), weather (Weather), economics (Exchange), and disease (ILI). The statistics of each dataset are summarized and found in Table 5. As a mainstream benchmark, the ETT datasets are extensively utilized for assessing long-term forecasting methods Zhou et al. (2021); Wu et al. (2021); Zhou et al. (2022b); Zeng et al. (2023); Wu et al. (2023). ETT consists of critical

---

[1]https://github.com/thuml/Time-Series-Library

Table 5: Statistics of nine datasets.

| Dataset | Domain | Time series length | The number of variables | Sampling frequency |
|---|---|---|---|---|
| ETTh1 | System Monitoring | 17420 | 7 | 1 Hour |
| ETTh2 | System Monitoring | 17420 | 7 | 1 Hour |
| ETTm1 | System Monitoring | 69680 | 7 | 15 minutes |
| ETTm2 | System Monitoring | 69680 | 7 | 15 minutes |
| Electricity | Energy Consumption | 26304 | 321 | 1 Hour |
| Traffic | Traffic | 17544 | 862 | 1 Hour |
| Weather | Weather | 52695 | 21 | 10 Minutes |
| Exchange | Economic | 7588 | 8 | 1 Day |
| ILI | Medical | 966 | 7 | 1 Week |

indicators (such as oil temperature, load, and others) that are gathered over a span of two years from electricity transformers. These datasets are grouped into four distinct sets based on location (ETT1 and ETT2) and time interval (15 minutes and one hour). The Electricity dataset captures the hourly electricity consumption of 321 customers from 2012 to 2014. On the other hand, the Traffic dataset compiles hourly data from the California Department of Transportation, detailing the occupancy rates of roads as measured by different sensors on freeways in the San Francisco Bay area. The Weather dataset consists of 21 meteorological indicators, including air temperature, humidity, and others, recorded at 10-minute intervals over the course of a year. The Exchange dataset chronicles daily exchange rates of eight different countries from 1990 through 2016. Lastly, the ILI dataset includes weekly records of influenza-like illness (ILI) patient data from the Centers for Disease Control and Prevention in the United States, spanning from 2002 to 2021. This dataset illustrates the ratio of patients diagnosed with ILI relative to the total patient count.

### A.4 BASELINE MODELS

In the realm of long-term forecasting, numerous models have been proposed since the advent of Informer. These models have demonstrated commendable performance with unique novelties. However, they were compared with underperforming models such as the models based on RNNs, and Transformer-based models, which are known to be susceptible to overfitting. Therefore, our primary focus is on high-performing and state-of-the-art models among the most recent proposals. We validate our method against seven forecasting baselines and three representation methodologies. All models were implemented using PyTorch. For the latest forecasting models, namely TimesNet[2], DLinear and NLinear[3], MICN[4], FiLM[5], Nonstationary Transformer[6], and FEDformer[7], we utilized the official code released by the original authors rather than implementing it from scratch.

Similarly, for recent representation methods, such as LaST[8], CoST[9], and TS2Vec[10], we utilized the official codes provided by the authors instead of implementing models from scratch. We adhered to the unique hyperparameters of each model, tuning within the parameter search range that yielded optimal performance. However, certain configurations, such as input length and output length, were set uniformly for ease of comparison. More specific evaluation protocols will be presented in the following section.

---

[2]https://github.com/thuml/Time-Series-Library
[3]https://github.com/cure-lab/LTSF-Linear
[4]https://github.com/wanghq21/MICN
[5]https://github.com/DAMO-DI-ML/NeurIPS2022-FiLM
[6]https://github.com/thuml/Nonstationary_Transformers
[7]https://github.com/MAZiqing/FEDformer
[8]https://github.com/zhycs/LaST
[9]https://github.com/salesforce/CoST
[10]https://github.com/yuezhihan/ts2vec

## A.5 EVALUATION DETAILS

In our experiments, we aim to assess the model's capability to capture long-term variations, so that the output length should be long enough to be affected by these variations. Increasing the output length more than those used in previous experiments, however, involves several considerations. Thus, we delineate the modifications to the standard evaluation protocol as follows:

1. The input length $I$ is set to 14 (for the ILI dataset), 48 (for the Exchange dataset), 192 (for the ETTm dataset), and 96 (for the others datasets). These input lengths allow us to increase the output length within the limited window length, in accordance with the total length of each dataset.

2. The standard protocol splits all datasets into training, validation, and test sets in chronological order, with a ratio of 6:2:2 for the ETT dataset and 7:1:2 for the remaining datasets. However, due to the increased window length in the other datasets, the validation set is insufficiently populated. Therefore, we adopt a ratio of 6:2:2 for all datasets.

3. The weather dataset contains negative values for indicators that should logically be non-negative. These erroneous labels, if not corrected, could impede accurate evaluation due to scaling issues. We rectified these errors by filling them with neighboring values.

We adhered to standard protocols for all experiments, barring the exceptions outlined above.

## A.6 HYPERPARAMETER SENSITIVITY

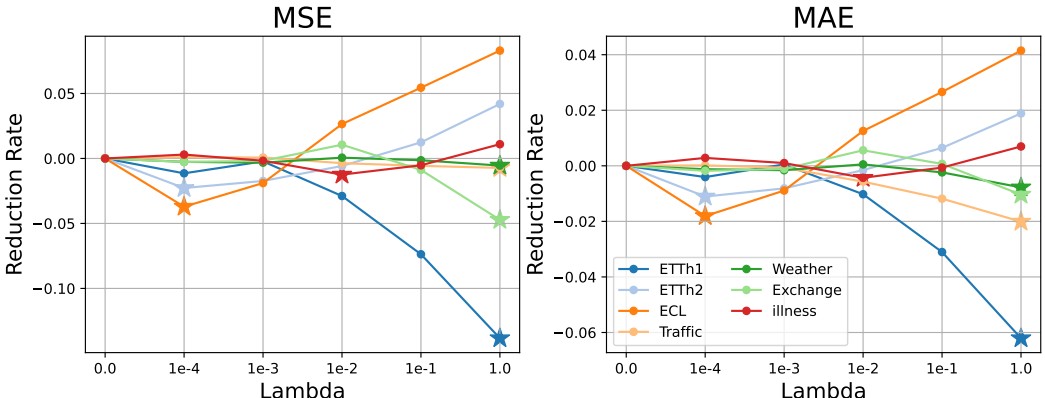

Figure 9: Error reduction rates of seven datasets according to the lambda value of AutoCon: (Left) Mean squared error and (Right) Mean absolute error. The error reduction rates were calculated based on the lambda, which is set at 0.0 as the reference point. The best performances in all datasets were achieved when using our AutoCon (denoted the star markers). It is important to note that a high lambda value does not necessarily imply strong long-term variations.

# B ADDITIONAL EXPERIMENTS

## B.1 EXPERIMENTS ON WINDOW LENGTH

As mentioned in Section 5, we considered simply increasing the length of the window to capture the long-term variations as much as possible. Also, as the window length increases, the number of data windows used for learning decreases. After all, we hypothesize that increasing the window length increases the input complexity of the model, while reducing the number of data points, making the model vulnerable to overfitting.

Figures 10 and 11 depict the training and testing losses when the input length increase from 192 to 920 for the purpose of predicting 720 steps in ETTh1 and ETTh2, respectively. We observed that the overall test loss tends to soar or converge, while the training loss persistently decreases when the input size is increased in five models with varying capacities and attributes.

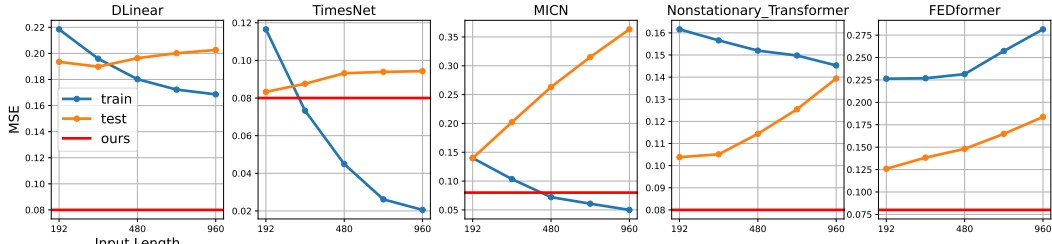

Figure 10: The figure shows the training (blue line) and test (orange line) loss trace plots of five models as the input length gradually increases on the ETTh1 with the output-720 setting. In the figure, the red horizontal line represents the performance of our model with the input-96 setting.

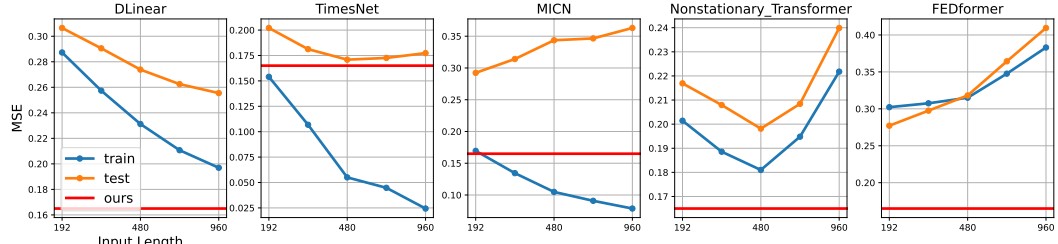

Figure 11: The figure shows the training (blue line) and test (orange line) loss trace plots of five models as the input length gradually increases on the ETTh2 with the output-720 setting. In the figure, the red horizontal line represents the performance of our model with the input-96 setting.

Also, in the case of DLinear in Figure 11, both test and training losses decline in sync due to limited capacity. However, we regard it as an underfitting problem since test errors are higher than our method (see red line). Consequently, we empirically substantiate that merely increasing the input length does not necessarily enhance the performance of long-term forecasting. Moreover, it is noteworthy that the computational cost for complex models other than linear models increases significantly with increasing sequence length.

## B.2 ADDITIONAL FIGURE 6 RESULTS ON OTHER DATASETS

Additionally, we provide results for ETTh1 (Figure 12) and Electricity (Figure 13), which show the long-term variations. Although there are some differences in magnitude, the overall trends are similar across the three datasets.

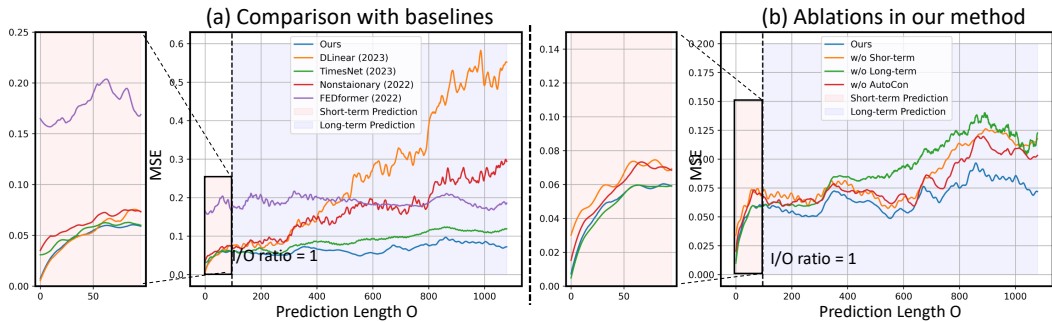

Figure 12: Comparison of forecasting error (MSE) according to prediction length $O$. (a) comparison with baseline models and (b) comparison between ablations of our method on the ETTh1 dataset.

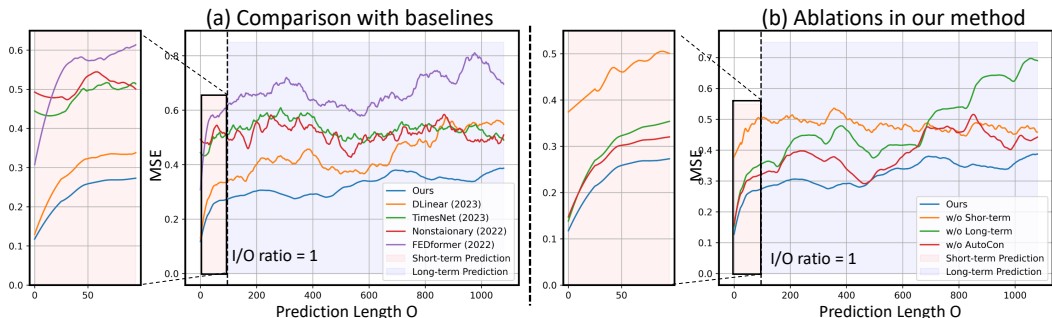

Figure 13: Comparison of forecasting error (MSE) according to prediction length $O$. (a) comparison with baseline models and (b) comparison between ablations of our method on the Electricity dataset.

Table 6: Increasing complexity of long-term branch in DLinear.

| Model | # of layers | ETTh1 | | ETTh2 | | Electricity | |
|---|---|---|---|---|---|---|---|
| | | MSE | MAE | MSE | MAE | MSE | MAE |
| DLinear | 1 | **0.1780±0.0054** | **0.3466±0.0063** | **0.2929±0.0140** | **0.4362±0.0087** | 0.3067±0.01544 | 0.4125±0.0109 |
| | 2 | 0.1993±0.1964 | 0.3638±0.2191 | 0.3170±0.0488 | 0.4581±0.0404 | 0.3775±0.01574 | 0.4618±0.0081 |
| | 3 | 0.2793±0.0678 | 0.4594±0.0682 | 0.3008±0.0046 | 0.4451±0.0035 | **0.3057±0.00902** | **0.4097±0.0065** |
| | 4 | 0.2760±0.0663 | 0.4564±0.0683 | 0.3015±0.0031 | 0.4455±0.0024 | 0.3165±0.03838 | 0.4183±0.0279 |

## B.3 ADDITIONAL ABLATION RESULTS FOR LONG-TERM BRANCH

Increasing the complexity of the long-term branch is essential for learning long-term representations, but it is not the sole reason for the superiority of our methodology. In other words, even with the increased complexity, capturing long-term variation is not easy in the current framework that uses only forecasting loss. As the main contribution, using AutoCon is essential for learning long-term variation and leading to performance improvement. To verify this, we additionally present two ablation results: increasing complexity in DLinear and in our model.

First, DLinear utilizes only a single linear layer for both long-term and short-term branches. We increase the complexity of the long-term branch by stacking linear layers with an activation function in the long-term branch. However, as evident in Table 6 below, even when stacking layers in the long term, performance tends to decrease or remain similar. This demonstrates that increasing long-term complexity is not effective in the existing decomposition architecture.

Second, the following Table 7 demonstrates the performance changes based on the complexity of the long-term branch in our decomposition architecture. Without Autocon, our model may be slightly better or comparable to the second-best model. The highest performance is achieved only when AutoCon is employed. This further underscores the necessity of the AutoCon we proposed to accurately predict long-term variation.

## B.4 ANALYSIS OF COMPUTATIONAL COST

Given the real-time nature of most time-series applications, computational efficiency is a crucial factor in time-series forecasting (Dannecker, 2015; Iqbal et al., 2019; Torres et al., 2021). As forecasting horizons increase, the window length expands, leading to increased computational costs. Therefore, it is imperative to evaluate a model's computational efficiency. Figure 14 illustrates the time required to update the parameters by a single batch for our model in comparison with the baseline models. The computational cost was measured for four different output lengths, ranging from 96 to 2160, for each model. A batch size of 32 was used, and all measurements were taken independently in the same GPU and server environment. Firstly, the linear model took the least amount of time due to its minimal number of parameters and the simplicity of matrix multiplication operations. On the other hand, TimesNet required the most time as it extracts multiple periods and computes a loop for each period. The Nonstationary model, which is based on the Transformer, has a computational complexity of $\mathcal{O}(W^2)$ in relation to length, which explains the sharp increase in computation time with length. Overall, our model was the second fastest after the linear model, and its computation cost did not

Table 7: Increasing complexity of long-term branch in our model.

| Model | # of layers | AutoCon | ETTh1 | |
|---|---|---|---|---|
| | | | MSE | MAE |
| TimesNet (baseline best) | 2 | - | 0.0834±0.0024 | 0.2310±0.0023 |
| Ours | 1 | X | 0.0837±0.0185 | 0.2372±0.0294 |
| Ours | 2 | X | 0.0910±0.0188 | 0.2360±0.0257 |
| Ours | 3 | X | 0.0876±0.0240 | 0.2351±0.0303 |
| Ours | 4 | X | 0.0918±0.0194 | 0.2381±0.0241 |
| Ours (best) | 1 | O | **0.0787±0.002** | **0.2226 ±0.0023** |

| Model | # of layers | AutoCon | ETTh2 | |
|---|---|---|---|---|
| | | | MSE | MAE |
| TimesNet (baseline best) | 2 | - | 0.2074±0.0113 | 0.3703±0.0155 |
| Ours | 1 | X | 0.2023±0.0230 | 0.3594±0.0261 |
| Ours | 2 | X | 0.2020±0.0098 | 0.3525±0.0078 |
| Ours | 3 | X | 0.2036±0.0265 | 0.3573±0.0224 |
| Ours | 4 | X | 0.2087±0.0167 | 0.3605±0.0130 |
| Ours (best) | 3 | O | **0.1771 ±0.0393** | **0.3441±0.0366** |

| Model | # of layers | AutoCon | Electricity | |
|---|---|---|---|---|
| | | | MSE | MAE |
| DLinear (baseline best) | 1 | - | 0.3067±0.0154 | 0.4125±0.0109 |
| Ours | 1 | X | 0.2928±0.1369 | 0.3978±0.1009 |
| Ours | 2 | X | 0.2889±0.0330 | 0.3927±0.0209 |
| Ours | 3 | X | 0.2975±0.0458 | 0.4049±0.0481 |
| Ours | 4 | X | 0.3089±0.4737 | 0.4115±0.0587 |
| Ours (best) | 2 | O | **0.2753±0.0224** | **0.3861±0.0166** |

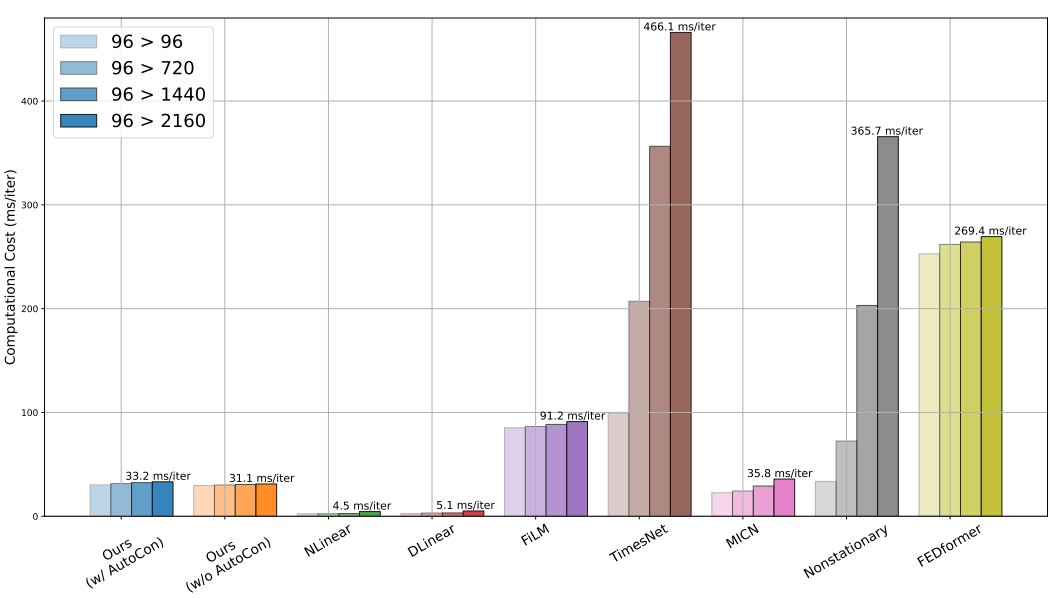

Figure 14: The figure illustrates the comparison of computational costs, measured in milliseconds per a single batch iteration (ms/iter), of the baseline models and our model. The computational cost for each model was evaluated by increasing the output length from 96 to 2160.

significantly increase even when training included AutoCon (from 31.1 ms/iter to 33.2 ms/iter). Hence, our method manages to achieve superior long-term forecasting performance compared to

linear models, while requiring less computational resources than other more complex models. The cost analysis was briefly addressed in Section 4.4 of the main paper.

## B.5 VISUALIZATION OF FORECASTING RESULTS

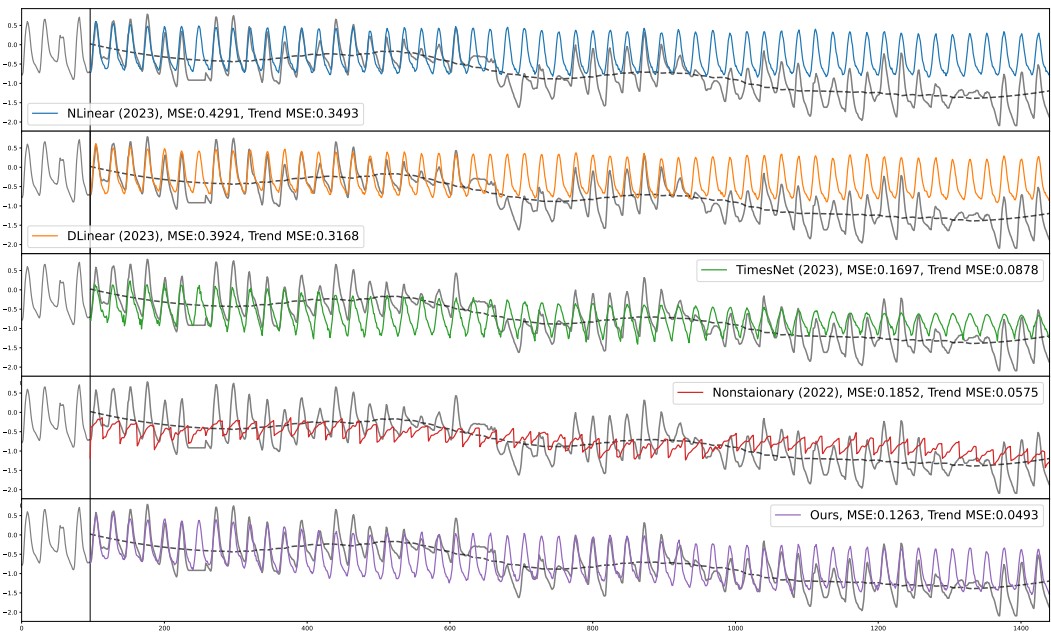

Figure 15: The figure displays the visualization of forecasting results for the 96-1440 setting in ETTh2, showcasing our model along with four other models.

Figure 15 provides a qualitative result of the five different models by visualizing the prediction results for 1440 steps in the ETTh2 dataset. In the case of linear models, the error increases as the prediction distance increases, failing to account for long-term variations. Nonstationary and TimesNet models, although better at following long-term variations than the linear model, struggle to capture high-frequency patterns effectively. Our model, on the other hand, successfully manages to capture both long-term variations and high-frequency patterns. This can be attributed to our model's structure, which is designed to benefit from both short-term and long-term predictions.

## B.6 EVALUATION RESULTS WITH OTHER METRICS

While existing metrics (*i.e.,* MSE and MAE) are standard metrics for long-term forecasting evaluation, they have limitations. Specifically, they may not adequately capture aspects such as the shape and temporal alignment of the time series, which are crucial for a comprehensive evaluation of a forecasting model's performance.

To address these limitations, we introduced two additional metrics based on Dynamic Time Warping (DTW) (Sakoe & Chiba, 1978): Shape DTW and Temporal DTW (Le Guen & Thome, 2019). Shape DTW focuses on the similarity of the pattern or shape of the predicted sequence to the actual sequence, providing insight into the model's ability to capture the underlying pattern of the time series. Temporal DTW evaluates the alignment of the predicted sequence with the actual sequence, highlighting the model's accuracy in forecasting the timing of events.

These additional metrics offer a more nuanced assessment of our model's performance, particularly in areas where MSE and MAE may fall short. Lower values in both Shape DTW and Temporal DTW indicate better performance, signifying lesser distortion between the predicted and actual sequences. As shown in Table 8, our method demonstrates superior performance not only in MSE and MAE but also in these shape and temporal alignment-focused metrics.

Table 8: Univariate forecasting results, evaluated using Shape and Temporal DTW in a 720-output setting across three datasets

| Dataset | ETTh1 | | ETTh2 | | Electricity | |
|---|---|---|---|---|---|---|
| Model \Metric | Shape DTW | Temporal DTW | Shape DTW | Temporal DTW | Shape DTW | Temporal DTW |
| Ours | **17.14±1.653** | **59.66±1.739** | **42.38±0.630** | **13.47±1.793** | **80.73±7.798** | **0.09±0.014** |
| TimesNet | 25.80±4.349 | 86.86±15.509 | 62.58±5.390 | 51.19±21.834 | 139.83±16.516 | 0.49±0.826 |
| PatchTST | 22.21±1.226 | 72.23±4.411 | 65.45±2.976 | 15.23±0.882 | 116.50±29.878 | 0.75±1.674 |
| MICN | 37.08±12.393 | 65.70±3.588 | 67.69±8.796 | 22.67±3.168 | 123.93±17.543 | 1.90±1.176 |
| DLinear | 58.32±1.955 | 155.21±8.587 | 82.53±3.627 | 24.88±2.403 | 88.70±3.550 | 0.18±0.145 |

## C  ANALYSIS OF REPRESENTATION

### C.1  DETAILS OF REPRESENTATION SIMILARITIES

In Figure 2, we used three baselines with our model and extracted the representations of each baseline either before the final projection layer (TimesNet) or after the encoder layer (PatchTST, FEDformer, and Ours). Our main purpose in visualizing the representation was to demonstrate the learning of long-term correlations. To display more clearly, we applied a filtering method to smooth short-term fluctuations within the given window. We also provide original representation results, which are enlarged for each baseline, without smoothing out the short-term fluctuations as shown in Figure 16. Figure 16 shows that the three baseline models learn the short-term correlations within the window, although they do not learn the long-term correlations.

One interesting point in this finding is that existing models have attempted to address the limitations of window length by leveraging time-stamp information. Actually, TimesNet, FEDformer, and our model obtained the representations using timestamps, incorporating them into the input sequences, while PatchTST does not utilize the timestamps. However, not only PatchTST but also both TimesNet and FEDformer do not effectively capture the annual cyclic patterns, despite utilizing timestamps as the same as our model. These failures are noteworthy, particularly considering the Electricity time series, which displays a yearly-long periodicity. These results show that it is challenging for the model to learn yearly patterns even when given input sequences and timestamps, solely relying on the existing forecasting loss. Therefore, this result demonstrates the necessity of our AutoCon loss. Furthermore, to justify the emergence of long-term representation irrespective of the model's structural aspects, we provide additional results of the representation from an ablation model that does not utilize AutoCon in our model. As shown in Figure 17, the model without AutoCon also exhibits a weak periodicity, but similar to other baselines, representation similarity remains relatively flat compared to our full model.

### C.2  VISUALIZATION OF REPRESENTATIONS

We benchmarked our AutoCon method against three representation learning methods proposed to enhance forecasting performance. TS2Vec and CoST have a two-stage learning framework in which they utilize a ridge regression model for time-series forecasting and deep learning-based models for representation learning. On the other hand, LaST and our method adopt an end-to-end learning framework wherein both representation and time-series forecasting learning occur concurrently.

Figure 7 presents the representation results of four methods over the ETTh2 dataset. To investigate whether each method has learned the representation structure associated with the long-term variations, we extracted representations corresponding to all training time steps and visualized them via UMAP with the month labels derived from the timestamp. Our model clearly displays continuity between adjacent months and demonstrates well-defined clustering, attributes not seen in the other models. It seems that the other models did not learn the structure necessary for recognizing one-year long-term variations beyond the window, as their representation learning was confined within the window.

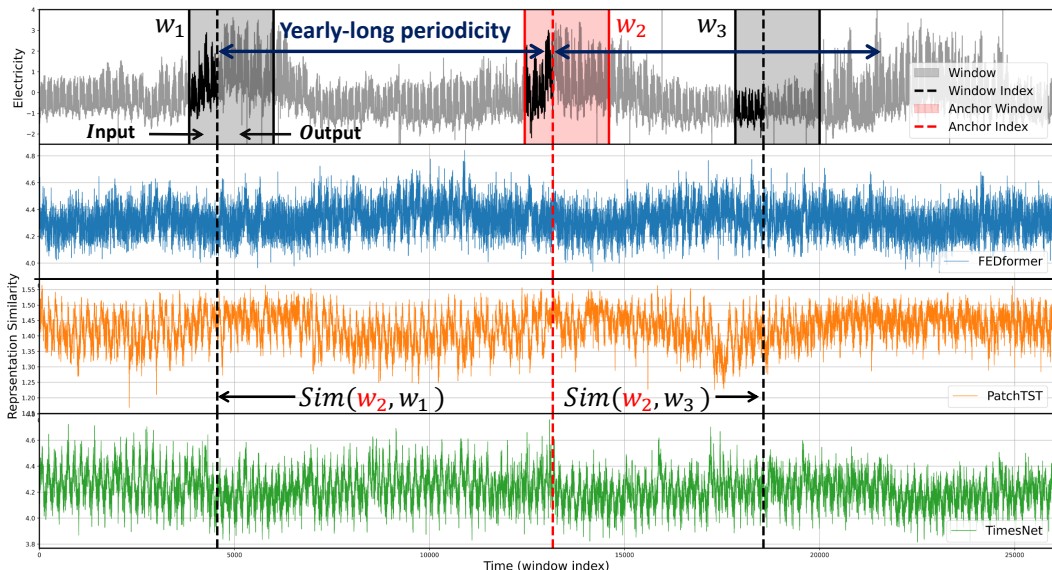

Figure 16: (Top) Electricity time series including a long-term variation beyond window size. (Bottom) Plotted representation similarities of three baseline models between an anchor window $\mathcal{W}_2$ and all other windows including $\mathcal{W}_1$ and $\mathcal{W}_3$.

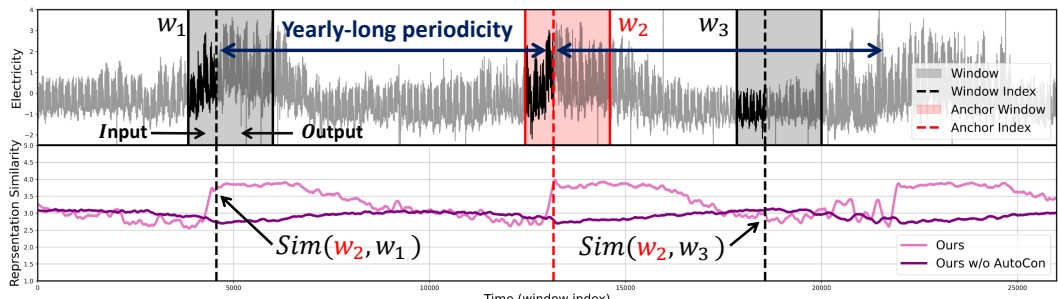

Figure 17: (Top) Electricity time series including a long-term variation beyond window size. (Bottom) Plotted representation similarities of our models (with AutoCon and without AutoCon) between an anchor window $\mathcal{W}_2$ and all other windows including $\mathcal{W}_1$ and $\mathcal{W}_3$.

### C.3 ADDITIONAL COMPARISON WITH TWO SELF-SUPERVISED LOSS

We designed and provided results for two possible self-supervised objectives based on HierCon (Yue et al., 2022) and SupCon (Khosla et al., 2020) that can be incorporated into our two-stream model structure. HierCon induces the representations of two partially overlapped windows to be close to each other, while SupCon encourages the encoder to learn close representations for the windows with the same month label.

The two SSL objectives were tested with our model architecture, replacing only the AutoCon loss. As shown in Table 9, compared to the one without any SSL loss, HierCon shows a slight improvement in performance on the ETTh1 and ETTh2 for short-term predictions with a length of 96, but it performs worse in the long-term prediction, as it emphasizes only temporal closeness. SupCon leveraged monthly information beyond the window length, leading to performance improvements even in the long-term prediction. However, SupCon can only learn a single predefined periodicity, unlike AutoCon. Consequently, SupCon shows lower performance than AutoCon in learning the periodicities existing in the time series through autocorrelation.

Table 9: Comparison with different self-supervised objectives in our redesigned architecture.

| Datasets | | ETTh1 | | | | ETTh2 | | | | Electricity | | | |
|---|---|---|---|---|---|---|---|---|---|---|---|---|---|
| Prediction length | | 96 | 720 | 1440 | 2160 | 96 | 720 | 1440 | 2160 | 96 | 720 | 1440 | 2160 |
| Ours w/o SSL | MSE | 0.061 | 0.082 | 0.096 | 0.130 | 0.147 | 0.214 | 0.213 | 0.236 | 0.206 | 0.289 | 0.363 | 0.419 |
| | MAE | 0.190 | 0.226 | 0.245 | 0.286 | 0.285 | 0.375 | 0.365 | 0.372 | 0.322 | 0.393 | 0.460 | 0.503 |
| Ours w/ HierCon | MSE | 0.059 | 0.090 | 0.121 | 0.145 | 0.132 | 0.221 | 0.221 | 0.263 | 0.223 | 0.313 | 0.380 | 0.500 |
| | MAE | 0.187 | 0.240 | 0.276 | 0.306 | 0.279 | 0.379 | 0.378 | 0.407 | 0.339 | 0.407 | 0.474 | 0.544 |
| Ours w/ SupCon | MSE | 0.056 | 0.082 | 0.092 | 0.091 | 0.125 | 0.185 | 0.199 | 0.215 | 0.209 | 0.279 | 0.351 | 0.408 |
| | MAE | 0.184 | 0.229 | 0.242 | 0.237 | 0.270 | 0.349 | 0.360 | 0.371 | 0.326 | 0.388 | 0.451 | 0.489 |
| Ours w/ AutoCon | MSE | **0.056** | **0.079** | **0.079** | **0.074** | **0.124** | **0.177** | **0.176** | **0.198** | **0.196** | **0.275** | **0.338** | **0.380** |
| | MAE | **0.182** | **0.223** | **0.225** | **0.215** | **0.269** | **0.344** | **0.340** | **0.358** | **0.313** | **0.386** | **0.441** | **0.481** |

Table 10: Multivariate forecasting results on ETT datasets with different prediction lengths $O \in \{96, 192, 336, 720\}$ with the input length $I = 96$. The full benchmark is available at Appendix D

| Models | | Ours | | TimesNet* (2023) | | MICN (2023) | | Crossformer (2023) | | N-HiTS (2023) | | DLinear* (2023) | | ETSformer* (2022) | | LightTS* (2022) | |
|---|---|---|---|---|---|---|---|---|---|---|---|---|---|---|---|---|---|
| | $O$ | MSE | MAE | MSE | MAE | MSE | MAE | MSE | MAE | MSE | MAE | MSE | MAE | MSE | MAE | MSE | MAE |
| ETTh1 | 96 | 0.387 | **0.396** | **0.384** | 0.402 | 0.421 | 0.431 | 0.407 | 0.429 | 0.404 | 0.424 | 0.386 | 0.400 | 0.494 | 0.479 | 0.424 | 0.432 |
| | 192 | 0.437 | **0.424** | **0.436** | 0.429 | 0.474 | 0.487 | 0.505 | 0.496 | 0.465 | 0.466 | 0.437 | 0.432 | 0.538 | 0.504 | 0.475 | 0.462 |
| | 336 | **0.476** | **0.442** | 0.491 | 0.469 | 0.569 | 0.551 | 0.620 | 0.574 | 0.519 | 0.501 | 0.481 | 0.459 | 0.574 | 0.521 | 0.518 | 0.488 |
| | 720 | **0.468** | **0.461** | 0.521 | 0.500 | 0.770 | 0.672 | 0.830 | 0.701 | 0.621 | 0.569 | 0.519 | 0.516 | 0.562 | 0.535 | 0.547 | 0.533 |
| Average | | **0.442** | **0.431** | 0.458 | 0.450 | 0.559 | 0.535 | 0.591 | 0.550 | 0.502 | 0.490 | 0.456 | 0.452 | 0.542 | 0.510 | 0.491 | 0.479 |
| ETTh2 | 96 | **0.290** | **0.341** | 0.340 | 0.374 | 0.299 | 0.364 | 0.645 | 0.562 | 0.346 | 0.381 | 0.333 | 0.387 | 0.340 | 0.391 | 0.397 | 0.437 |
| | 192 | **0.373** | **0.398** | 0.402 | 0.414 | 0.441 | 0.454 | 0.788 | 0.636 | 0.427 | 0.440 | 0.477 | 0.476 | 0.430 | 0.439 | 0.520 | 0.504 |
| | 336 | **0.408** | **0.423** | 0.452 | 0.452 | 0.654 | 0.567 | 0.959 | 0.500 | 0.518 | 0.500 | 0.594 | 0.541 | 0.485 | 0.479 | 0.626 | 0.559 |
| | 720 | **0.419** | **0.442** | 0.462 | 0.468 | 0.956 | 0.716 | 1.146 | 0.784 | 0.888 | 0.645 | 0.831 | 0.657 | 0.500 | 0.497 | 0.863 | 0.672 |
| Average | | **0.372** | **0.401** | 0.414 | 0.427 | 0.588 | 0.525 | 0.885 | 0.673 | 0.545 | 0.491 | 0.559 | 0.515 | 0.439 | 0.452 | 0.602 | 0.543 |
| ETTm1 | 96 | 0.330 | **0.365** | 0.338 | 0.375 | **0.316** | **0.362** | 0.364 | 0.399 | 0.355 | 0.389 | 0.345 | 0.372 | 0.375 | 0.398 | 0.374 | 0.400 |
| | 192 | 0.371 | **0.384** | 0.374 | 0.387 | **0.363** | 0.390 | 0.431 | 0.441 | 0.404 | 0.414 | 0.380 | 0.389 | 0.408 | 0.410 | 0.400 | 0.407 |
| | 336 | **0.399** | **0.408** | 0.410 | 0.411 | 0.408 | 0.426 | 0.517 | 0.488 | 0.452 | 0.456 | 0.413 | 0.413 | 0.435 | 0.428 | 0.438 | 0.438 |
| | 720 | **0.460** | **0.444** | 0.478 | 0.450 | 0.481 | 0.476 | 0.698 | 0.627 | 0.500 | 0.485 | 0.474 | 0.453 | 0.499 | 0.462 | 0.527 | 0.502 |
| Average | | **0.390** | **0.400** | 0.400 | 0.406 | 0.392 | 0.414 | 0.503 | 0.489 | 0.428 | 0.436 | 0.403 | 0.407 | 0.429 | 0.425 | 0.435 | 0.437 |
| ETTm2 | 96 | **0.178** | **0.260** | 0.187 | 0.267 | 0.179 | 0.275 | 0.272 | 0.357 | 0.201 | 0.287 | 0.193 | 0.292 | 0.189 | 0.280 | 0.209 | 0.308 |
| | 192 | **0.244** | **0.303** | 0.249 | 0.309 | 0.307 | 0.376 | 0.335 | 0.414 | 0.295 | 0.354 | 0.284 | 0.362 | 0.253 | 0.319 | 0.311 | 0.382 |
| | 336 | **0.305** | **0.341** | 0.321 | 0.351 | 0.325 | 0.388 | 0.564 | 0.590 | 0.359 | 0.391 | 0.369 | 0.427 | 0.314 | 0.357 | 0.442 | 0.466 |
| | 720 | **0.398** | **0.396** | 0.408 | 0.403 | 0.502 | 0.490 | 1.203 | 0.779 | 0.530 | 0.498 | 0.554 | 0.522 | 0.414 | 0.413 | 0.675 | 0.587 |
| Average | | **0.281** | **0.325** | 0.291 | 0.333 | 0.328 | 0.382 | 0.593 | 0.535 | 0.346 | 0.383 | 0.350 | 0.401 | 0.293 | 0.342 | 0.409 | 0.436 |

∗ denotes the results are taken from TimesNet.

# D  FULL BENCHMARKS

Table 10 and Table 11 show the full benchmark results including confidence intervals.

Table 11: Extended long-term forecasting full benchmarks with confidence interval. 'S'hort and 'L'ong indicate the length of the conventional experiment and the newly extended experiment setting, respectively. Nonstationary has an out-of-memory (OOM) problem at the output-4320 setting on ETTm1 and ETTm2 datasets.

| Models I→O | | | Ours | | TimesNet (2023) | | MICN (2023) | | PatchTST (2023) | | DLinear (2023) | | FiLM (2023) | | Nonstationary (2022) | | FEDformer (2022) | |
|---|---|---|---|---|---|---|---|---|---|---|---|---|---|---|---|---|---|---|
| | | | MSE | MAE | MSE | MAE | MSE | MAE | MSE | MAE | MSE | MAE | MSE | MAE | MSE | MAE | MSE | MAE |
| ETTh1 | 96 | S | **0.056** | **0.182** | 0.058 | 0.185 | 0.062 | 0.185 | 0.057 | 0.184 | 0.063 | 0.185 | 0.057 | **0.180** | 0.069 | 0.197 | 0.080 | 0.218 |
| | | | ±0.0015 | ±0.0020 | ±0.0014 | ±0.0028 | ±0.0020 | ±0.0028 | ±0.0017 | ±0.0038 | ±0.0038 | ±0.0052 | ±0.0006 | ±0.0010 | ±0.0068 | ±0.0099 | ±0.0037 | ±0.0039 |
| | 720 | | **0.079** | **0.223** | 0.083 | 0.230 | 0.175 | 0.342 | 0.089 | 0.236 | 0.180 | 0.348 | 0.097 | 0.245 | 0.117 | 0.272 | 0.130 | 0.285 |
| | | | ±0.0085 | ±0.0072 | ±0.0024 | ±0.0023 | ±0.0122 | ±0.0140 | ±0.0006 | ±0.0007 | ±0.0362 | ±0.0394 | ±0.0018 | ±0.0021 | ±0.0241 | ±0.0310 | ±0.0073 | ±0.0076 |
| | 1440 | L | **0.079** | **0.225** | 0.098 | 0.250 | 0.320 | 0.476 | 0.118 | 0.275 | 0.433 | 0.567 | 0.123 | 0.280 | 0.184 | 0.349 | 0.199 | 0.356 |
| | | | ±0.0120 | ±0.0158 | ±0.0051 | ±0.0066 | ±0.0476 | ±0.0436 | ±0.0041 | ±0.0045 | ±0.2573 | ±0.2044 | ±0.0019 | ±0.0019 | ±0.0119 | ±0.0139 | ±0.0174 | ±0.0193 |
| | 2160 | | **0.074** | **0.215** | 0.143 | 0.303 | 0.539 | 0.637 | 0.183 | 0.355 | 0.629 | 0.698 | 0.187 | 0.359 | 0.334 | 0.504 | 0.356 | 0.486 |
| | | | ±0.0108 | ±0.0166 | ±0.0310 | ±0.0373 | ±0.0157 | ±0.0107 | ±0.0102 | ±0.0116 | ±0.1734 | ±0.1050 | ±0.0032 | ±0.0039 | ±0.0996 | ±0.0985 | ±0.0439 | ±0.0235 |
| ETTh2 | 96 | S | 0.124 | 0.269 | 0.139 | 0.290 | **0.122** | **0.264** | 0.136 | 0.292 | 0.128 | 0.271 | 0.129 | 0.275 | 0.192 | 0.343 | 0.129 | 0.277 |
| | | | ±0.0043 | ±0.0056 | ±0.0030 | ±0.0036 | ±0.0013 | ±0.0010 | ±0.0019 | ±0.0018 | ±0.0008 | ±0.0007 | ±0.0014 | ±0.0021 | ±0.0198 | ±0.0177 | ±0.0046 | ±0.0066 |
| | 720 | | **0.177** | **0.344** | 0.207 | 0.370 | 0.313 | 0.457 | 0.233 | 0.392 | 0.319 | 0.461 | 0.256 | 0.407 | 0.231 | 0.394 | 0.273 | 0.419 |
| | | | ±0.0393 | ±0.0366 | ±0.0113 | ±0.0155 | ±0.0048 | ±0.0039 | ±0.0046 | ±0.0042 | ±0.0215 | ±0.0170 | ±0.0041 | ±0.0036 | ±0.0155 | ±0.0121 | ±0.0132 | ±0.0101 |
| | 1440 | L | **0.176** | **0.340** | 0.192 | 0.358 | 0.520 | 0.599 | 0.351 | 0.481 | 0.514 | 0.597 | 0.389 | 0.506 | 0.211 | 0.379 | 0.384 | 0.487 |
| | | | ±0.0042 | ±0.0046 | ±0.0574 | ±0.0621 | ±0.1879 | ±0.1216 | ±0.0172 | ±0.0130 | ±0.0591 | ±0.0413 | ±0.0081 | ±0.0052 | ±0.0165 | ±0.0155 | ±0.0327 | ±0.0225 |
| | 2160 | | **0.198** | **0.358** | 0.263 | 0.413 | 0.759 | 0.734 | 0.610 | 0.659 | 0.740 | 0.728 | 0.610 | 0.645 | 0.240 | 0.399 | 0.919 | 0.737 |
| | | | ±0.0367 | ±0.0307 | ±0.1384 | ±0.1105 | ±0.0469 | ±0.0255 | ±0.0552 | ±0.0355 | ±0.0932 | ±0.0515 | ±0.0120 | ±0.0063 | ±0.0284 | ±0.0237 | ±0.1815 | ±0.0698 |
| ETTm1 | 192 | S | 0.042 | 0.157 | 0.044 | 0.161 | 0.045 | 0.160 | **0.039** | **0.150** | 0.045 | 0.156 | 0.041 | 0.154 | 0.058 | 0.181 | 0.066 | 0.200 |
| | | | ±0.0017 | ±0.0061 | ±0.0011 | ±0.0014 | ±0.0032 | ±0.0060 | ±0.0003 | ±0.0005 | ±0.0040 | ±0.0062 | ±0.0001 | ±0.0003 | ±0.0052 | ±0.0079 | ±0.0240 | ±0.0402 |
| | 1440 | | 0.090 | 0.238 | **0.085** | **0.227** | 0.099 | 0.245 | 0.091 | 0.237 | 0.105 | 0.252 | 0.098 | 0.247 | 0.142 | 0.298 | 0.093 | 0.238 |
| | | | ±0.0133 | ±0.0258 | ±0.0019 | ±0.0022 | ±0.0058 | ±0.0060 | ±0.0052 | ±0.0076 | ±0.0024 | ±0.0028 | ±0.0003 | ±0.0002 | ±0.0340 | ±0.0350 | ±0.0014 | ±0.0013 |
| | 2880 | L | **0.089** | **0.238** | 0.092 | 0.242 | 0.137 | 0.294 | 0.096 | 0.245 | 0.175 | 0.344 | 0.101 | 0.251 | 0.181 | 0.339 | 0.127 | 0.282 |
| | | | ±0.0052 | ±0.0066 | ±0.0056 | ±0.0071 | ±0.0213 | ±0.0252 | ±0.0054 | ±0.0075 | ±0.0098 | ±0.0110 | ±0.0010 | ±0.0011 | ±0.1038 | ±0.1067 | ±0.0068 | ±0.0083 |
| | 4320 | | **0.083** | **0.228** | 0.090 | 0.241 | 0.247 | 0.412 | 0.110 | 0.265 | 0.271 | 0.437 | 0.119 | 0.275 | OOM | OOM | 0.167 | 0.324 |
| | | | ±0.0094 | ±0.0097 | ±0.0033 | ±0.0038 | ±0.0516 | ±0.0538 | ±0.0048 | ±0.0057 | ±0.0111 | ±0.0114 | ±0.0003 | ±0.0003 | | | ±0.0113 | ±0.0083 |
| ETTm2 | 192 | S | **0.093** | **0.227** | 0.102 | 0.240 | 0.095 | 0.230 | 0.094 | 0.231 | 0.093 | 0.230 | 0.096 | 0.232 | 0.121 | 0.260 | 0.110 | 0.257 |
| | | | ±0.0019 | ±0.0027 | ±0.0019 | ±0.0028 | ±0.0022 | ±0.0025 | ±0.0015 | ±0.0018 | ±0.0001 | ±0.0001 | ±0.0009 | ±0.0015 | ±0.0100 | ±0.0084 | ±0.0140 | ±0.0184 |
| | 1440 | | **0.215** | **0.362** | 0.228 | 0.378 | 0.243 | 0.388 | 0.226 | 0.380 | 0.237 | 0.384 | 0.235 | 0.386 | 0.280 | 0.424 | 0.264 | 0.408 |
| | | | ±0.0072 | ±0.0055 | ±0.0073 | ±0.0067 | ±0.0168 | ±0.0123 | ±0.0102 | ±0.0103 | ±0.0018 | ±0.0010 | ±0.0013 | ±0.0026 | ±0.0231 | ±0.0152 | ±0.0265 | ±0.0212 |
| | 2880 | L | **0.211** | **0.370** | 0.236 | 0.391 | 0.322 | 0.465 | 0.243 | 0.396 | 0.322 | 0.464 | 0.262 | 0.412 | 0.268 | 0.417 | 0.302 | 0.441 |
| | | | ±0.0188 | ±0.0170 | ±0.0097 | ±0.0103 | ±0.0088 | ±0.0066 | ±0.0158 | ±0.0136 | ±0.0016 | ±0.0014 | ±0.0023 | ±0.0020 | ±0.0452 | ±0.0297 | ±0.0143 | ±0.0108 |
| | 4320 | | **0.215** | **0.376** | 0.234 | 0.393 | 0.448 | 0.555 | 0.307 | 0.450 | 0.448 | 0.553 | 0.333 | 0.468 | OOM | OOM | 0.409 | 0.518 |
| | | | ±0.0242 | ±0.0228 | ±0.0295 | ±0.0249 | ±0.0335 | ±0.0236 | ±0.0157 | ±0.0100 | ±0.0182 | ±0.0135 | ±0.0047 | ±0.0030 | | | ±0.0312 | ±0.0191 |
| Electricity | 96 | S | **0.196** | **0.313** | 0.286 | 0.386 | 0.241 | 0.367 | 0.227 | 0.336 | 0.207 | 0.322 | 0.394 | 0.451 | 0.332 | 0.426 | 0.279 | 0.393 |
| | | | ±0.0024 | ±0.0043 | ±0.0286 | ±0.0188 | ±0.0053 | ±0.0086 | ±0.0142 | ±0.0102 | ±0.0025 | ±0.0021 | ±0.0019 | ±0.0021 | ±0.0341 | ±0.0239 | ±0.0160 | ±0.0102 |
| | 720 | | **0.275** | **0.386** | 0.417 | 0.471 | 0.336 | 0.446 | 0.332 | 0.426 | 0.304 | 0.412 | 0.467 | 0.504 | 0.505 | 0.533 | 0.417 | 0.486 |
| | | | ±0.0471 | ±0.0426 | ±0.0270 | ±0.0149 | ±0.0732 | ±0.0592 | ±0.0015 | ±0.0005 | ±0.0077 | ±0.0064 | ±0.0032 | ±0.0016 | ±0.1046 | ±0.0581 | ±0.0319 | ±0.0221 |
| | 1440 | L | **0.338** | **0.441** | 0.491 | 0.523 | 0.419 | 0.504 | 0.482 | 0.537 | 0.395 | 0.484 | 0.625 | 0.610 | 0.577 | 0.574 | 0.651 | 0.609 |
| | | | ±0.0256 | ±0.0047 | ±0.0245 | ±0.0117 | ±0.0346 | ±0.0258 | ±0.0046 | ±0.0093 | ±0.0150 | ±0.0117 | ±0.0028 | ±0.0010 | ±0.0591 | ±0.0248 | ±0.0470 | ±0.0237 |
| | 2160 | | **0.380** | **0.481** | 0.536 | 0.547 | 0.421 | 0.501 | 0.768 | 0.644 | 0.415 | 0.496 | 0.938 | 0.758 | 0.642 | 0.610 | 0.896 | 0.714 |
| | | | ±0.0307 | ±0.0199 | ±0.0526 | ±0.0272 | ±0.0190 | ±0.0137 | ±0.1452 | ±0.0578 | ±0.0098 | ±0.0094 | ±0.0039 | ±0.0024 | ±0.1659 | ±0.0910 | ±0.1156 | ±0.0529 |
| Traffic | 96 | S | **0.132** | **0.206** | 0.145 | 0.219 | 0.168 | 0.256 | 0.192 | 0.296 | 0.219 | 0.327 | 0.264 | 0.334 | 0.247 | 0.326 | 0.220 | 0.312 |
| | | | ±0.0014 | ±0.0008 | ±0.0006 | ±0.0013 | ±0.0115 | ±0.0113 | ±0.0033 | ±0.0020 | ±0.0013 | ±0.0016 | ±0.0018 | ±0.0021 | ±0.0099 | ±0.0086 | ±0.0224 | ±0.0184 |
| | 720 | | **0.144** | **0.225** | 0.163 | 0.269 | 0.304 | 0.394 | 0.213 | 0.318 | 0.309 | 0.419 | 0.247 | 0.329 | 0.277 | 0.360 | 0.255 | 0.344 |
| | | | ±0.0005 | ±0.0006 | ±0.0006 | ±0.0002 | ±0.0130 | ±0.0124 | ±0.0085 | ±0.0079 | ±0.0002 | ±0.0002 | ±0.0015 | ±0.0014 | ±0.0243 | ±0.0235 | ±0.0649 | ±0.0546 |
| | 1440 | L | **0.174** | **0.251** | 0.188 | 0.292 | 0.375 | 0.443 | 0.246 | 0.341 | 0.353 | 0.409 | 0.311 | 0.390 | 0.303 | 0.361 | 0.297 | 0.376 |
| | | | ±0.0009 | ±0.0016 | ±0.0002 | ±0.0219 | ±0.0301 | ±0.0250 | ±0.0147 | ±0.0143 | ±0.0108 | ±0.0083 | ±0.0022 | ±0.0025 | ±0.0327 | ±0.0204 | ±0.0446 | ±0.0358 |
| | 2160 | | **0.175** | **0.252** | 0.190 | 0.304 | 0.360 | 0.426 | 0.261 | 0.353 | 0.324 | 0.386 | 0.988 | 0.745 | 0.222 | 0.317 | 0.317 | 0.394 |
| | | | ±0.0074 | ±0.0082 | ±0.0006 | ±0.0171 | ±0.0257 | ±0.0043 | ±0.0078 | ±0.0053 | ±0.0078 | ±0.0061 | ±0.0033 | ±0.0021 | ±0.0208 | ±0.0288 | ±0.0431 | ±0.0419 |
| Weather | 96 | S | **0.521** | **0.522** | 0.584 | 0.536 | 0.569 | 0.525 | 0.545 | 0.539 | 0.579 | 0.529 | 0.589 | 0.533 | 0.636 | 0.567 | 0.703 | 0.625 |
| | | | ±0.0522 | ±0.0582 | ±0.0114 | ±0.0060 | ±0.0102 | ±0.0040 | ±0.0012 | ±0.0014 | ±0.0074 | ±0.0029 | ±0.0034 | ±0.0017 | ±0.0227 | ±0.0035 | ±0.1516 | ±0.0994 |
| | 720 | | **0.963** | **0.715** | 1.090 | 0.753 | 1.080 | 0.754 | 0.987 | 0.752 | 1.007 | 0.706 | 1.003 | 0.728 | 1.007 | 0.725 | 1.114 | 0.822 |
| | | | ±0.0193 | ±0.0076 | ±0.0297 | ±0.0083 | ±0.0500 | ±0.0195 | ±0.0133 | ±0.0051 | ±0.0293 | ±0.0101 | ±0.0046 | ±0.0017 | ±0.0350 | ±0.0127 | ±0.0325 | ±0.0163 |
| | 1440 | L | **1.280** | **0.835** | 1.547 | 0.926 | 1.351 | 0.863 | 1.342 | 0.860 | 1.299 | 0.823 | 1.472 | 0.900 | 1.394 | 0.867 | 1.435 | 0.919 |
| | | | ±0.1150 | ±0.0400 | ±0.3281 | ±0.0858 | ±0.0199 | ±0.0075 | ±0.0160 | ±0.0046 | ±0.0248 | ±0.0091 | ±0.0030 | ±0.0011 | ±0.0730 | ±0.0272 | ±0.1262 | ±0.0537 |
| | 2160 | | **1.415** | **0.887** | 1.744 | 0.994 | 1.544 | 0.937 | 1.506 | 0.924 | 1.454 | 0.887 | 1.712 | 0.988 | 1.598 | 0.944 | 1.786 | 1.054 |
| | | | ±0.1449 | ±0.0501 | ±0.1810 | ±0.0737 | ±0.0482 | ±0.0185 | ±0.0333 | ±0.0109 | ±0.0160 | ±0.0059 | ±0.0027 | ±0.0011 | ±0.2034 | ±0.0783 | ±0.3309 | ±0.1121 |
| Exchange | 48 | S | 0.051 | 0.172 | 0.054 | 0.178 | 0.054 | 0.181 | 0.068 | 0.197 | **0.049** | **0.170** | 0.052 | 0.173 | 0.054 | 0.178 | 0.059 | 0.184 |
| | | | ±0.0007 | ±0.0019 | ±0.0010 | ±0.0019 | ±0.0007 | ±0.0010 | ±0.0037 | ±0.0059 | ±0.0005 | ±0.0011 | ±0.0005 | ±0.0011 | ±0.0017 | ±0.0038 | ±0.0030 | ±0.0041 |
| | 360 | | **0.448** | **0.527** | 0.479 | 0.532 | 0.459 | 0.536 | 0.548 | 0.573 | 0.485 | 0.531 | 0.492 | 0.534 | 0.493 | 0.541 | 0.528 | 0.556 |
| | | | ±0.0579 | ±0.0141 | ±0.0097 | ±0.0062 | ±0.0496 | ±0.0208 | ±0.0051 | ±0.0056 | ±0.0808 | ±0.0561 | ±0.0057 | ±0.0034 | ±0.0948 | ±0.0407 | ±0.0206 | ±0.0082 |
| | 720 | L | **1.067** | **0.794** | 1.239 | 0.856 | 1.383 | 0.927 | 1.264 | 0.859 | 1.718 | 1.024 | 1.291 | 0.864 | 1.358 | 0.894 | 1.381 | 0.903 |
| | | | ±0.4944 | ±0.1552 | ±0.0556 | ±0.0224 | ±0.1069 | ±0.0390 | ±0.0369 | ±0.0128 | ±0.5115 | ±0.1862 | ±0.0167 | ±0.0038 | ±0.0793 | ±0.0291 | ±0.1234 | ±0.0415 |
| | 1080 | | **1.004** | **0.792** | 1.327 | 0.900 | 4.874 | 1.972 | 1.255 | 0.873 | 4.982 | 1.973 | 1.670 | 1.010 | 1.774 | 1.058 | 1.600 | 0.980 |
| | | | ±0.2767 | ±0.0853 | ±0.0578 | ±0.0198 | ±0.2948 | ±0.0696 | ±0.0269 | ±0.0098 | ±0.9254 | ±0.1950 | ±0.0821 | ±0.0227 | ±1.1516 | ±0.3573 | ±0.1925 | ±0.0597 |
| ILI | 14 | S | **0.725** | **0.574** | 1.414 | 0.735 | 0.815 | 0.701 | 1.558 | 0.965 | 1.397 | 0.901 | 1.079 | 0.739 | 1.107 | 0.698 | 0.773 | 0.619 |
| | | | ±0.0955 | ±0.0467 | ±0.2329 | ±0.0204 | ±0.0157 | ±0.0088 | ±0.1390 | ±0.0428 | ±0.1129 | ±0.0420 | ±0.0264 | ±0.0126 | ±0.1107 | ±0.0399 | ±0.0546 | ±0.0341 |
| | 28 | | **0.887** | **0.683** | 1.604 | 0.854 | 1.670 | 1.062 | 1.878 | 1.110 | 2.008 | 1.134 | 1.315 | 0.887 | 1.515 | 0.767 | 0.989 | 0.770 |
| | | | ±0.0756 | ±0.0212 | ±0.1568 | ±0.0301 | ±0.0739 | ±0.0274 | ±0.0419 | ±0.0116 | ±0.1461 | ±0.0468 | ±0.0776 | ±0.0385 | ±0.1455 | ±0.0301 | ±0.0787 | ±0.0450 |
| | 56 | L | **0.807** | **0.725** | 1.021 | 0.787 | 1.757 | 1.210 | 1.451 | 1.028 | 1.584 | 1.075 | 1.080 | 0.891 | 0.895 | 0.742 | 0.856 | 0.741 |
| | | | ±0.0113 | ±0.0118 | ±0.0673 | ±0.0307 | ±0.0401 | ±0.0129 | ±0.0570 | ±0.0150 | ±0.0874 | ±0.0300 | ±0.0221 | ±0.0122 | ±0.0668 | ±0.0550 | ±0.0487 | ±0.0286 |
| | 112 | | **1.499** | **1.038** | 1.669 | 1.072 | 3.593 | 1.759 | 2.846 | 1.438 | 3.332 | 1.572 | 2.608 | 1.387 | 1.724 | 1.108 | 1.660 | 1.097 |
| | | | ±0.0908 | ±0.0335 | ±0.1123 | ±0.0462 | ±0.0515 | ±0.0127 | ±0.0670 | ±0.0159 | ±0.0821 | ±0.0145 | ±0.2469 | ±0.0801 | ±0.2964 | ±0.1245 | ±0.1745 | ±0.0560 |
| 1st Count | | | 52 | | 2 | | 2 | | 2 | | 4 | | 1 | | 0 | | 0 | |

