# OpenReview forum: "Self-Supervised Contrastive Learning for Long-term Forecasting"
_ICLR.cc/2024/Conference — ICLR 2024 poster_

### Official Review · Reviewer_hiP7 · 2023-10-28

**Soundness:** 3 good
**Presentation:** 3 good
**Contribution:** 2 fair
**Rating:** 6
**Confidence:** 4

**Summary:**

This paper addresses challenges of long-term forecasting by utilizes contrastive learning and an enhanced decomposition architecture specifically designed to address long-term variations. The key idea considers the global autocorrelation within the entire time series, enabling the creation of positive and negative pairs in a self-supervised manner. Experiments demonstrate that this approach outperforms baseline models on nine established long-term forecasting benchmarks.

**Strengths:**

Contribution:

- The authors deliver an intuitive and easy to implement technique based on newly defined ‘global autocorrelation’. A plus point for this technique also relates to its fast computing time and no extensive memory requirement.
- The method is extensible and applicable to both uni and multi-variate datasets.
- Well-rounded experiments are conducted with multiple datasets of different patterns (different autocorrelation patterns - Fig. 5).
- The authors are well aware with the weakness of autocorrelation technique - which capture only linear variations.

Presentation:

- Nice visual explainations (e.g. Fig.1, 3).

**Weaknesses:**

- For circumstance of nearly stationary input sequence, the autocorrelation will produce stationary result and cause the collapse of Constrastive Learning framework.
- The increasing complexity for long-term branch of decomposition framework might contribute to the better result of the whole proposed pipeline. A potential verification could be an additional ablation case of that architecture with a linear layer like the work of Zeng (2023).
- While the authors are aware of the linear assumption of autocorrelation and suggest the potential use of different techniques for high-order one, this suggestions might be inapplicable for the lengthy global input sequences with densely use frequency. Any approaches investigating these non-linear correlation can consume much more resources compare to the current autocorrelation.
- Some grammar and typos (e.g. TiemsNet - page 2)

Reference:

Ailing Zeng, Muxi Chen, Lei Zhang, and Qiang Xu. Are transformers effective for time series
forecasting? In Proc. the AAAI Conference on Artificial Intelligence (AAAI), 2023

**Questions:**

Please address my comments on weaknesses.

---

> ### Author Response · Authors · 2023-11-20
>
> ### W1: Collapse of our contrastive framework with stationary time-series
> We thank reviewer hiP7 for the insightful comment regarding the potential challenges posed by nearly stationary input sequences in our contrastive learning framework.
> We understand your concern that "the nearly stationary sequences" might lead to stationary autocorrelation results, which could, in turn, affect the contrastive learning mechanism.
>
> However, even in the nearly stationary sequences, the autocorrelation does not produce stationary results since stationary results in autocorrelation are theoretically possible only in the case of infinitely long time series.
> In practice, we deal with finite samples from time series. In such cases, autocorrelation converges to zero, decreasing from 1 to 0 as the lag $\tau$ increases even in "the nearly stationary time series". In other words, autocorrelation exhibits a non-stationary trend and shows a decreasing pattern. This is because the covariance with lagged series $Cov(S_{t+\tau}, S_t) $ decreases as $\tau$ increases when numerically calculating the autocorrelation.
> For example, as shown in M-Figure 5, the autocorrelation results of four datasets converge regardless of their stationarity.
>
> Additionally, the Weather time series, which is considered the most stationary among the tested time series according to the Augmented Dick-Fuller (ADF) test [[1]](https://www.nber.org/papers/t0130) as shown in statistics [[2]](https://proceedings.neurips.cc/paper_files/paper/2022/hash/4054556fcaa934b0bf76da52cf4f92cb-Abstract-Conference.html), also converges to 0 in a similar manner.
> Naturally, such stationarity implies small (or absence of) long-term variation.
> Therefore, even if our method is applied, as described in Dataset Analysis of Section 4.1, performance improvement could be marginal (3% reduced MSE).
> However, we believe there is an unequivocal difference between gaining only a slight performance improvement and "collapse of contrastive learning".
> (If reviewer hiP7 was trying to convey a different meaning than "collapse", we would be happy to further discuss this matter.)
>
> ### W2: Increasing complexity of long-term branch in the decomposition architecture
> Increasing the complexity of the long-term branch is essential for learning long-term representations, but it is not the only reason for the superior performance of our methodology.
> In other words, even with the increased complexity, capturing long-term variation is not an easy task if the current framework only uses the forecasting loss.
> It is essential to use the AutoCon loss to effectively learn long-term variation that leads to performance improvement.
> We have already demonstrated the need for AutoCon loss through ablations in M-Figure 6 and M-Table 3.
> However, to fully address the reviewer's concerns, we conducted two additional ablation studies: increasing complexity in DLinear [[3]](https://ojs.aaai.org/index.php/AAAI/article/view/26317) and in our model.
>
> First, DLinear utilizes only a single linear layer for both long-term and short-term branches.
> We increase the complexity of the long-term branch by stacking linear layers with an activation function.
> However, as evident in R-Table 3 below, even when stacking layers in the long-term branch, performance tends to decrease or remain similar.
> This demonstrates that merely increasing the complexity of the long-term branch is not effective in the existing decomposition architecture.
>
> **R-Table 3: Increasing Complexity of Long-term Branch in DLinear in input-336-output-720 univariate setting**
>
> | **Model** | **\# of Layers** | **ETTh1 (MSE)**   | **ETTh1 (MAE)**   | **ETTh2 (MSE)**   | **ETTh2 (MAE)**   | **Electricity (MSE)** | **Electricity (MAE)** |
> |------------|-------------------|-------------------|-------------------|-------------------|-------------------|-----------------------|-----------------------|
> | DLinear    | 1                 | **0.1780±0.0054** | **0.3466±0.0063** | **0.2929±0.0140** | **0.4362±0.0087** | 0.3067±0.01544        | 0.4125±0.0109         |
> |            | 2                 | 0.1993±0.1964     | 0.3638±0.2191     | 0.3170±0.0488     | 0.4581±0.0404     | 0.3775±0.01574        | 0.4618±0.0081         |
> |            | 3                 | 0.2793±0.0678     | 0.4594±0.0682     | 0.3008±0.0046     | 0.4451±0.0035     | **0.3057±0.00902**    | **0.4097±0.0065**     |
> |            | 4                 | 0.2760±0.0663     | 0.4564±0.0683     | 0.3015±0.0031     | 0.4455±0.0024     | 0.3165±0.03838        | 0.4183±0.0279         |
>
> * In the case of the 2 layers, there exist linear layers along with the temporal axis only (*i.e.,* the outputs shapes as follows: (B, I, C) > (B, D, C) > (B, O, C) where B is batch size, I is input length, D is hidden dimension, and O is output length)

---

> ### Author Response · Authors · 2023-11-20
>
> Second, the following R-Table 4 demonstrates the change in performance based on the complexity of the long-term branch in our decomposition architecture. Without Autocon, our model may be slightly better or comparable to the second-best model.
> The highest performance is achieved only when the AutoCon loss is employed. This further underscores the necessity of the AutoCon loss we proposed in order to accurately capture long-term variation.
>
> **R-Table 4: Increasing complexity of long-term branch of our model over three datasets in output-720 univariate setting. Other setting is the same to experiments as shown in M-Table 1.**
>
> | **Model**                 | **\# of layers** | **AutoCon** | **ETTh1 (MSE)**  | **ETTh1 (MAE)**   |
> |---------------------------|-------------------|-------------|------------------|-------------------|
> | TimesNet (baseline best)  | 2                 | -           | 0.0834±0.0024    | 0.2310±0.0023     |
> | Ours                      | 1                 | X           | 0.0837±0.0185    | 0.2372±0.0294     |
> | Ours                      | 2                 | X           | 0.0910±0.0188    | 0.2360±0.0257     |
> | Ours                      | 3                 | X           | 0.0876±0.0240    | 0.2351±0.0303     |
> | Ours                      | 4                 | X           | 0.0918±0.0194    | 0.2381±0.0241     |
> | Ours (best)               | 1                 | O           | **0.0787±0.002** | **0.2226±0.0023** |
>
>
> | **Model**                 | **\# of layers** | **AutoCon** | **ETTh2 (MSE)**   | **ETTh2 (MAE)**   |
> |---------------------------|-------------------|-------------|-------------------|-------------------|
> | TimesNet (baseline best)  | 2                 | -           | 0.2074±0.0113     | 0.3703±0.0155     |
> | Ours                      | 1                 | X           | 0.2023±0.0230     | 0.3594±0.0261     |
> | Ours                      | 2                 | X           | 0.2020±0.0098     | 0.3525±0.0078     |
> | Ours                      | 3                 | X           | 0.2036±0.0265     | 0.3573±0.0224     |
> | Ours                      | 4                 | X           | 0.2087±0.0167     | 0.3605±0.0130     |
> | Ours (best)               | 3                 | O           | **0.1771±0.0393** | **0.3441±0.0366** |
>
>
>
> | **Model**                 | **\# of layers** | **AutoCon** | **Electricity (MSE)** | **Electricity (MAE)** |
> |---------------------------|-------------------|-------------|-----------------------|-----------------------|
> | DLinear (baseline best)   | 1                 | -           | 0.3067±0.0154         | 0.4125±0.0109         |
> | Ours                      | 1                 | X           | 0.2928±0.1369         | 0.3978±0.1009         |
> | Ours                      | 2                 | X           | 0.2889±0.0330         | 0.3927±0.0209         |
> | Ours                      | 3                 | X           | 0.2975±0.0458         | 0.4049±0.0481         |
> | Ours                      | 4                 | X           | 0.3089±0.4737         | 0.4115±0.0587         |
> | Ours (best)               | 2                 | O           | **0.2753±0.0224**     | **0.3861±0.0166**     |
>
> ### W3: Applying our AutoCon with high-order and non-linear correlation
>
> As reviewer hiP7 mentioned, thoroughly calculating high-order correlation across a very long time series can be computationally challenging.
> What we have claimed and demonstrated in this work, however, is the necessity of leveraging global autocorrelation for long-term predictions.
> These two are independent topics, and we find it difficult to understand why the former is pointed out as a "weakness" of the latter.
> Even if one decides to combine the two somehow in the future (*e.g.* extend our work to incorporate high-order global correlation), there are many possibilities to explore. For example, one could try to reduce the computational complexity by leveraging domain knowledge of the specific dataset or the task (*e.g.* ignore the correlation between certain variables or certain time durations).
> Alternatively, one could propose to capture the global high-order correlation in a completely different manner than how we capture the global autocorrelation.
> Therefore, we find it unconvincing that the reviewer would use an independent problem as a weakness for the current submission.
>
>
> ### W4: Typos
> Thank the reviewer for pointing out the typos. We have corrected them in the revised manuscript.
>
> ### References
> [[1]](https://www.nber.org/papers/t0130) Efficient tests for an autoregressive unit root, 1992
> [[2]](https://proceedings.neurips.cc/paper_files/paper/2022/hash/4054556fcaa934b0bf76da52cf4f92cb-Abstract-Conference.html) Non-stationary Transformers: Exploring the Stationarity in Time Series Forecasting, NeurIPS'22
> [[3]](https://ojs.aaai.org/index.php/AAAI/article/view/26317) Are Transformers Effective for Time Series Forecasting?, AAAI'23

---

> > ### Comment · Reviewer_hiP7 · 2023-11-21
> > **Additional Comment from Reviewer hiP7**
> >
> > Dear Authors,
> >
> > Thank you for making clarifications to my concerns.
> > I can tell the Authors have made ample clarifications for my two first concerns.
> >
> > Regarding the third one, I agree that *leveraging global autocorrelation for long-term predictions* and *calculating high-order correlation* is two independent topics. However, what I want to clarify what I meant is that the way the Authors use linear autocorrelation for now in this work might limit the scopes of applications in the circumstance of that non-linear correlations represent.
> >
> > In general, I quite satisfy with the answer, together with the extra experiment the authors conduct.
> > I have updated my scores.
> >
> > Best,

---

> > > ### Author Response · Authors · 2023-11-22
> > >
> > > Dear Reviewer hiP7,
> > >
> > > Thank you for taking the time to review our responses.  We are pleased to hear that you are satisfied with our responses to the first two concerns.
> > >
> > > As for the third concern, we appreciate your insights into the potential limitations of utilizing non-linear and high-order correlations for long-term predictions. We acknowledge the distinction between using non-linear correlations and using linear correlation, especially concerning computational costs. We will carefully consider this aspect in future work to broaden the scope of applications.
> > >
> > > Best regards,

---

### Official Review · Reviewer_MDze · 2023-11-01

**Soundness:** 3 good
**Presentation:** 3 good
**Contribution:** 2 fair
**Rating:** 5
**Confidence:** 5

**Summary:**

This paper points out existing approaches fail to capture the long-term variations that are partially caught within the short window. Based on the finding, this paper presents a novel approach for long-term time series forecasting by employing contrastive learning and an enhanced decomposition architecture. The contrastive loss incorporates global autocorrelation held in the whole time series, which facilitates the construction of positive and negative pairs in a self-supervised manner. The authors conducted extensive experiments on nine benchmarks and achieved superior performance compared to baseline models.

**Strengths:**

* It is a good observation that long-term forecasting data exists long-term autocorrelations and existing methods fail to take into account.

* By adding autocorrelation constrained contrastive loss, it reaches long-term variation consistency between windows, making it more explainable.

* Extensive experiment results and ablation studies show the effectiveness of the method.

**Weaknesses:**

* Capturing long-term variation via the contrastive loss part is separated from the forecasting mechanism. Good finding but have no further design for architectures based on this finding. The forecasting module is a simple dual-head forecaster and already achieves good performance.

* Lack of performance comparisons of different self-supervised objectives.

* 14 baselines comparison is claimed in the abstract, only 7 are found.The performance improvement seems marginal. Extending the forecasting window too much has little practical meaning.

**Questions:**

Could you provide training protocols and the default value of the contrastive loss weight hyperparameter?

---

> ### Author Response · Authors · 2023-11-20
>
> ### W1: The forecasting module is a simple dual-head forecaster and already achieves good performance
>
> In response to the reviewer's concern that "The forecasting module is a simple dual-head forecaster and already achieves good performance", we have already demonstrated through an ablation study that the performance decreases when AutoCon is not used as shown in M-Table 3 and (b) of M-Figure 6.
> Therefore, without our contribution, the AutoCon loss, the dual-head forecaster fails to achieve good performance in long-term forecasting.
>
> And the reviewer claimed, "Capturing long-term variation via the contrastive loss part is separated from the forecasting mechanism", but our long-term encoder is jointly optimized by both forecasting and the AutoCon loss.
> This learning approach aligns with our initial proposal of the two-stream architecture, where the linear structure empowers short-term (motivated by recent linear models [[1]](https://ojs.aaai.org/index.php/AAAI/article/view/26317)), and the contrastive loss to capture the long-term variations.
> Empirically, considering the performance decline observed in the ablation studies when excluding the AutoCon loss, we find it unconvincing that reviewer MDze claims the contrastive loss operates independently from the forecasting mechanism.
>
> Finally, our deliberate model design has indeed demonstrated superior long-term prediction performance compared to baseline models.
> Therefore, as desired by the reviewer, we can incorporate even more complex architectures based on these findings in future work.

---

> ### Author Response · Authors · 2023-11-20
>
> ### W2:Lack of performance comparisons of different self-supervised objectives
> Since we used self-supervised learning (SSL) loss in addition to the forecasting loss, it seems reasonable to compare our methodology with approaches utilizing different self-supervised objectives.
> We fully agree with this point and have already presented comparisons with three recent methodologies that employ various self-supervised losses in Section 4.3. However, to comprehensively address reviewer MDze's concern, we have designed and provided results for two possible self-supervised objectives based on HierCon [[2]](https://ojs.aaai.org/index.php/AAAI/article/view/20881) and SupCon [[3]](https://proceedings.neurips.cc/paper/2020/hash/d89a66c7c80a29b1bdbab0f2a1a94af8-Abstract.html) that can be incorporated into our two-stream model structure.
> HierCon induces the representations of two partially overlapped windows to be close to each other, while SupCon encourages the encoder to learn close representations for the windows with the same month label.
>
> The two SSL objectives were tested with our model architecture, replacing only the AutoCon loss. As shown in R-Table 2 (we also added this in Appendix C.3), compared to the one without any SSL loss, HierCon shows a slight improvement in performance on the ETTh1 and ETTh2 for short-term predictions with a length of 96, but it performs worse in the long-term prediction, as it emphasizes only temporal closeness.
> SupCon leveraged monthly information beyond the window length, leading to performance improvements even in the long-term prediction.
> However, SupCon can only learn a single predefined periodicity, unlike AutoCon.
> Consequently, SupCon shows lower performance than AutoCon in learning the periodicities existing in the time series through autocorrelation.
> We additionally demonstrated the performance of well-known SSL baselines that are compatible with our model architecture to address the concerns of reviewers as much as possible.
> However, if there are specific SSL methods that reviewer MDze wishes to see, we would appreciate any suggestions or proposals from reviewer MDze.
>
> **R-Table 2: Comparison with different self-supervised objectives in our redesigned architecture. Another setting is the same to the experiments as shown in M-Table 1**
>
> |                     | Dataset | ETTh1     |  |  |  | ETTh2     |  |  |  | Electricity |           |           |           |
> |---------------------|---------|-----------| --- | --- | --- |-----------| --- | --- | --- |-------------|-----------|-----------|-----------|
> | **Model**           | **Output**  | **96**        | **720** | **1440** | **2160** | **96**        | **720** | **1440** | **2160** | **96**          | **720**       | **1440**      | **2160**      |
> | **Ours w/o SSL**    | **MSE** | 0.061     | 0.082 | 0.096 | 0.130 | 0.147     | 0.214 | 0.213 | 0.236 | 0.206       | 0.289     | 0.363     | 0.419     |
> |                     | **MAE** | 0.190     | 0.226 | 0.245 | 0.286 | 0.285     | 0.375 | 0.365 | 0.372 | 0.322       | 0.393     | 0.460     | 0.503     |
> | **Ours w/ HierCon** | **MSE** | 0.059     | 0.090 | 0.121 | 0.145 | 0.132     | 0.221 | 0.221 | 0.263 | 0.223       | 0.313     | 0.380     | 0.500     |
> |                     | **MAE** | 0.187     | 0.240 | 0.276 | 0.306 | 0.279     | 0.379 | 0.378 | 0.407 | 0.339       | 0.407     | 0.474     | 0.544     |
> | **Ours w/ SupCon**  | **MSE** | 0.056     | 0.082 | 0.092 | 0.091 | 0.125     | 0.185 | 0.199 | 0.215 | 0.209       | 0.279     | 0.351     | 0.408     |
> |                     | **MAE** | 0.184     | 0.229 | 0.242 | 0.237 | 0.270     | 0.349 | 0.360 | 0.371 | 0.326       | 0.388     | 0.451     | 0.489     |
> | **Ours w/ AutoCon** | **MSE** | **0.056** | **0.079** | **0.079** | **0.074** | **0.124** | **0.177** | **0.176** | **0.198** | **0.196**   | **0.275** | **0.338** | **0.380** |
> |                     | **MAE** | **0.182** | **0.223** | **0.225** | **0.215** | **0.269** | **0.344** | **0.340** | **0.358** | **0.313**   | **0.386** | **0.441** | **0.481** |

---

> ### Author Response · Authors · 2023-11-20
>
> ### W3:Experiment setting and results
> * 1: 14 baselines, but seven are found
> In our study, we analyzed 14 distinct baseline models, which are distributed across different sections of our results due to the breadth of our analysis.
> Specifically, we evaluated seven models in the context of univariate forecasting, as detailed in M-Table 1.
> Additionally, four models were examined in the multivariate forecasting domain, as shown in M-Table 2, which includes models such as Crossformer and N-HITS that are specifically tailored to learning inter-variable dependencies.
> Furthermore, we included three time-series representation methods in our analysis, the results of which are presented in M-Table 4.
> We apologize for causing confusion by stating the aggregate number as 14 in the abstract. In the revised version, we added the clue ``in multiple experiments'' to the abstract to prevent confusion.
>
> * 2: marginal performance improvement
> We understand that the reviewer's concerns about marginal performance improvement primarily stem from short-term prediction (Output-96) and the Weather dataset.
> However, our method contributes to capturing long-term variations beyond the window scope by leveraging global autocorrelation, thereby enhancing the performance of long-term predictions.
> Therefore, marginal performance improvement in the short-term setting is reasonably anticipated.
> In relation to the Weather dataset, we already elaborated in Dataset Analysis of Section 4.1 on how performance improvement varies based on the degree of long-term variation in the time series.
> In that analysis, we have already demonstrated that weather exhibits less long-term variation compared to other datasets, and we have explained that this could inevitably limit the performance improvement of our model. This analysis, rather, serves as significant evidence that our model is indeed operating as intended, validating our intuition that focused on the importance of long-term autocorrelation. In other words, it confirms that our model has behaved according to our intentions, affirming the accuracy of our emphasis on global autocorrelation.
> Most importantly, we believe stating "Performance improvement seems marginal" based on only a few numbers does not accurately capture the overall statistics.
> As shown in M-Table 1, our method achieved the top performance 42 times across all benchmarks.
> On average, it achieved a 12\% reduction in MSE and a maximum reduction of 34\% in MSE.
>
>
> * 3: practical meaning of extending window
> Contemporary time-series communities continue to propose numerous models and methods aimed at predicting the more distant future, with a focus on improving long-term performance.
> For instance, Informer [[4]](https://ojs.aaai.org/index.php/AAAI/article/view/17325), in response to the real-world need for long-term forecasting, extended the forecasting length beyond the conventional 48 output length—primarily evaluated in M4 [[5]](https://www.sciencedirect.com/science/article/pii/S0169207018300785?casa_token=d5VexiwRFBAAAAAA:pGKzZoJWfqMrB8ktzblyinqxUnZVjLu3JodaoHPihDy522ZTlhk1Hq2eYPiL2CraUhOepO60iHA);
> further expanded it to 720. To achieve both performance and computational efficiency in predicting a sequence of length 720, a huge number of models and methods have been proposed [[6](https://proceedings.neurips.cc/paper/2021/hash/bcc0d400288793e8bdcd7c19a8ac0c2b-Abstract.html), [7](https://proceedings.mlr.press/v162/zhou22g.html), [8](https://arxiv.org/abs/2210.02186), [9](https://arxiv.org/abs/2211.14730), [10](https://openreview.net/forum?id=zt53IDUR1U),].
> At the same time, ReVIN [[11]](https://openreview.net/forum?id=cGDAkQo1C0p) has successfully extended the prediction length up to 960, demonstrating that longer predictions become feasible when addressing and resolving the issue of distribution shift.
> Most recently, a new concurrent effort [[12]](https://openreview.net/forum?id=y08bkEtNBK) has emerged with the aim of further increasing the output length up to 2880.
> Given the collective efforts of numerous researchers, extending our input to 2160 is not merely a setting to demonstrate superiority in performance but is rooted in genuine practical significance and necessity.
> If reviewer MDze, however, can present a concrete argument as to why all these efforts have little practical meaning, we are more than happy to further discuss this matter.
> Long-term prediction is our core research interest, and if there is any convincing counter-argument we were unaware of, we would like to take it into account in defining our future research.

---

> ### Author Response · Authors · 2023-11-20
>
> ### Q1:Implementation details
> * 1:Training protocols
> Our training protocol is identical to conventional training methods, except for the inclusion of AutoCon as an additional loss, in addition to the forecasting loss. To clarify this further, we also present the [algorithm](https://tinyurl.com/training-algorithm) and added it to the Appendix A.1 of revision.
>
> * 2:Weight hyperparameter of AutoCon
> We already have shown the analysis of the optimal values and sensitivity for the contrastive loss weight hyperparameter for each dataset. However, due to the page limit, we included it in M-Figure 9 of Appendix A.6. We would appreciate it if the reviewer could refer to the appendix once again.
>
>
> ### References
> [[1]](https://ojs.aaai.org/index.php/AAAI/article/view/26317) Are Transformers Effective for Time Series Forecasting?, AAAI'23
> [[2]](https://ojs.aaai.org/index.php/AAAI/article/view/20881) TS2Vec: Towards Universal Representation of Time Series, AAAI'22
> [[3]](https://proceedings.neurips.cc/paper/2020/hash/d89a66c7c80a29b1bdbab0f2a1a94af8-Abstract.html) Supervised Contrastive Learning, NeurIPS'20
> [[4]](https://ojs.aaai.org/index.php/AAAI/article/view/17325) Informer: Beyond efficient transformer for long sequence time-series forecasting, AAAI'21
> [[5]](https://www.sciencedirect.com/science/article/pii/S0169207018300785?casa_token=d5VexiwRFBAAAAAA:pGKzZoJWfqMrB8ktzblyinqxUnZVjLu3JodaoHPihDy522ZTlhk1Hq2eYPiL2CraUhOepO60iHA) The M4 Competition: Results, findings, conclusion and way forward, International Journal of Forecasting'18
> [[6]](https://proceedings.neurips.cc/paper/2021/hash/bcc0d400288793e8bdcd7c19a8ac0c2b-Abstract.html) Autoformer: Decomposition transformers with auto-correlation for long-term series forecasting, NeurIPS'21
> [[7]](https://proceedings.mlr.press/v162/zhou22g.html) FEDformer: Frequency enhanced decomposed transformer for long-term series forecasting, ICML'22
> [[8]](https://arxiv.org/abs/2210.02186) TimesNet: Temporal 2D-Variation Modeling for General Time Series Analysis, ICLR'23
> [[9]](https://arxiv.org/abs/2211.14730) A Time Series is Worth 64 Words: Long-term Forecasting with Transformers, ICLR'23
> [[10]](https://openreview.net/forum?id=zt53IDUR1U) MICN: Multi-scale Local and Global Context Modeling for Long-term Series Forecasting, ICLR'23
> [[11]](https://openreview.net/forum?id=cGDAkQo1C0p) Reversible Instance Normalization for Accurate Time-Series Forecasting against Distribution Shift, ICLR22
> [[12]](https://openreview.net/forum?id=y08bkEtNBK) WITRAN: Water-wave Information Transmission and Recurrent Acceleration Network for Long-range Time Series Forecasting, NeurIPS'23

---

### Official Review · Reviewer_eBaK · 2023-11-02

**Soundness:** 3 good
**Presentation:** 3 good
**Contribution:** 3 good
**Rating:** 6
**Confidence:** 4

**Summary:**

This paper mainly presents a new contrastive objective in self-supervised representation learning in time series forecasting. Specifically, the paper argues that the trends extracted from a moving window are often long-term seasonal variations that cannot be captured by the window of smaller size. Therefore, the representations of windows within a mini-batch that are more similar are explicitly induced to get closer compared to other samples with lower correlations. The proposed contrastive loss is then added to a decomposition-based forecasting model as a regularization term. Experiments show impressive performance improvements over existing SOTA models.

**Strengths:**

1. Overall the paper is well-written and easy to follow. The idea of global autocorrelation and decomposition is well-motivated and sufficiently grounded, and the argument that short-term trends are long-term variations is quite convincing.
2. The empirical results on univariate long-term forecasting is very impressive.
3. Extensive analysis support the effectiveness of the proposed method with abundant evidences

**Weaknesses:**

1. Most of the ambiguity comes from the global information in the proposed representation. The authors should elaborate on the details of the encoder to highlight how the similarity of representations $v_i, v_j$ differ from the linear autocorrelation $r(i, j)$ in order to justify the contrastive objective (3). See question (1).
2. The plots of baseline models in figure 2 are not quite convincing. While the baseline models do not consider long-term correlations, it's counter-intuitive that the similarity of moving windows with various patterns stays almost constant as shown in the chart.

**Questions:**

1. How is the timestamp-based features derived and incorporated into the encoder?
2. Which dataset is used in Figure 6? Do you observe the same result on other datasets?

---

> ### Author Response · Authors · 2023-11-20
>
> ### W1:Details of the encoder to understand representation similarity
> We sincerely apologize for any ambiguity caused by an insufficient explanation. We mainly focused on experiments and visualization results to enhance understanding, which unfortunately led us to neglect the encoder details that should have been included for clarity. Following the reviewer eBaK's helpful guidance, we first clarify how the timestamp-based features are derived and incorporated into the encoder (Question 1).
>
> * Q1: How are the timestamp-based features derived and incorporated into the encoder?
> The timestamp signifies the moment at which data is observed, and this moment is expressed in units of 'year,' 'month,' 'day,' and 'hour.'
> The timestamp provides a forecasting model with periodicity on a yearly, monthly, and daily basis as input, aiding the model in learning various business cycles.
> Most models use timestamps for long-term forecasting, commonly adding numerical values of periodic functions aligned with each unit's cycle from the timestamp as features to the input sequence.
> Specifically, if we use four units (*i.e.,* month, day, weekday, and hour), we get a timestamp-based feature $f_{time} \in \mathbb{R}^{I \times 4} $, which is indexed by $\mathcal{T}$ and is aligned with input sequence $\mathcal{X} \in \mathbb{R}^{I \times c}$ where $I$ denote the input sequence length and $c$ is the number of variables.
> Then, with a projection layer $W_{time}$ and $b_{time}$, the time embedding $e_{time} = W_{time} \cdot f_{time} + b_{time} \in \mathbb{R}^{I \times d}$ is added to value embedding $e_{value} = W_{value}\cdot \mathcal{X}+b_{value}$.
> Finally, the temporal encoder (*e.g.,* RNN, TCN, and Attention)  takes the embedding sequence $e = e_{value} + e_{time} \in \mathbb{R}^{I \times d}$ to learn temporal dependency.
> This approach of utilizing the timestamp information was initially introduced in Informer [[1]](https://ojs.aaai.org/index.php/AAAI/article/view/17325), and our model, along with recent models [[2](https://proceedings.neurips.cc/paper/2021/hash/bcc0d400288793e8bdcd7c19a8ac0c2b-Abstract.html), [3](https://proceedings.mlr.press/v162/zhou22g.html), [4](https://arxiv.org/abs/2210.02186)], adopts the same approach.
>
> Now, we clarify the rationale for our AutoCon objective in relation to this timestamp information. In the baselines used in M-Figure 2, TimesNet, FEDformer, and our model all obtained representations using these timestamps, incorporating them into the input sequences, while PatchTST does not utilize the timestamps.
> An important observation is that both TimesNet and FEDformer do not effectively capture the annual cyclic patterns, despite utilizing timestamps as the same as our model.
> These failures are noteworthy, particularly considering the Electricity time series, which displays a yearly-long periodicity.
> These results show that it is challenging for the model to learn yearly patterns even when given input sequences and timestamps, solely relying on the existing forecasting loss.
> Therefore, this result demonstrates the necessity of our AutoCon loss.
> Furthermore, to better demonstrate the importance of our loss in capturing long-term periodicities, we conducted an additional experiment where we compared two models: Ours with and without AutoCon loss.
> As shown in [R-Figure1](https://tinyurl.com/wo-autocon-repr-sim), compared to the one with AutoCon loss, the one without AutoCon loss demonstrates only a hint of periodicity (not too different from other baselines).
>
>
> ### W2:Constant similarity of baseline models even with various patterns
> As pointed out by reviewer eBaK, M-Figure 2 indeed seems to suggest that the baselines failed to learn even the short-term correlations, not to mention long-term correlations.
> This is actually due to the smoothing we applied to the representation similarity, in order to help the readers better understand that none of the baselines are able to capture long-term correlations.
> We apologize for any confusion caused by not explicitly mentioning this process and have denoted it in the caption of M-Figure 2 in the revision.
> We also provide original representation results, which are enlarged for each baseline, without smoothing out the short-term fluctuations as shown in [R-Figure2](https://tinyurl.com/baselines-repr-sim).
> As anticipated by reviewer eBaK, the three baseline models are indeed learning short-term correlations within the window, although they are not learning long-term correlations.
> We hope that additional visualizations clarify the ambiguity caused by the smoothed representation similarity plots.

---

> ### Author Response · Authors · 2023-11-20
>
> ### Q2:Additional M-Figure 6 results over other datasets
> We thank reviewer eBaK for pointing out and addressing our oversight. M-Figure 6 presents the experimental results for the ETTh2 dataset. We explicitly identified it in the description of M-Figure 6 of our revised paper. Additionally, we provide results for ETTh1 ([R-Figure3](https://tinyurl.com/figure6-ETTh1)) and Electricity ([R-Figure4](https://tinyurl.com/figure6-ecl)), which show the long-term variations. Although there are some differences in magnitude, the overall trends are similar across the three datasets. We have also added these additional results to the Appendix B.2. We sincerely appreciate the reviewer`s constructive and helpful comments once again.
>
> ### References
> [[1]](https://ojs.aaai.org/index.php/AAAI/article/view/17325) Informer: Beyond efficient transformer for long sequence time-series forecasting, AAAI'21
> [[2]](https://proceedings.neurips.cc/paper/2021/hash/bcc0d400288793e8bdcd7c19a8ac0c2b-Abstract.html) Autoformer: Decomposition transformers with auto-correlation for long-term series forecasting, NeurIPS'21
> [[3]](https://proceedings.mlr.press/v162/zhou22g.html) FEDformer: Frequency enhanced decomposed transformer for long-term series forecasting, ICML'22
> [[4]](https://arxiv.org/abs/2210.02186) TimesNet: Temporal 2D-Variation Modeling for General Time Series Analysis, ICLR'23

---

### Official Review · Reviewer_pLfj · 2023-11-07

**Soundness:** 3 good
**Presentation:** 3 good
**Contribution:** 3 good
**Rating:** 8
**Confidence:** 4

**Summary:**

The paper is about self-supervised contrastive forecasting in long sequences. Existing methods perform poorly on capturing long-term variations beyond the window (outer-window variations). The proposed method, AutoCon, learns a long-term representation by constructing positive and negative pairs accross distant windows in a self-supervised manner. The authors have perform extensive experiments on 9 datasets, using 14 state of the art models, and MSE and MAE evaluation metrics to show that the models with AutoCon loss outperform the rest in most cases, achieving performance improvements of up to 34%.

**Strengths:**

- The paper is well-written and well-structured. The authors have done a great job with providing the related work and the limitations, giving the details to the proposed methodology and an extensive experimental evaluation. The images and tables are well-designed and show attention to detail.
- The paper is about an interesting problem, long-term time-series forecasting via contrastive learning. This is topic that has gained a lot of attention recently.
- The methodology is described with enough details.
- The extensive experiments, not only capture different cases in a various datasets and with various state of the art comparison methods, but they also show that the proposed methodology outperforms in most cases the other models.

**Weaknesses:**

- The novelty of the work is incemental. While the authors focus on an interesting problem, the proposed methodology is combination of existing components.
- It would be useful to add a table with the dataset statistics summarized (in Appendix if it does not fit in the main paper). How does the proposed methodology performs in imbalanced data?
- It would be interesting and more convincing on the performance if the authors would add results on more evaluation metrics, e.g., Accuracy, AUROC, AUPRC, etc.
- It would be interesting to show what is the runtime for each method/dataset and time complexities.

**Questions:**

- There is no discussion about making the code available publicly.
- The authors can respond on my comments 2-4 in the weaknesses section above.

**Details Of Ethics Concerns:**

No ethic concerns.

---

> ### Author Response · Authors · 2023-11-20
>
> ### W1: Novelty of the work
> We appreciate reviewer pLfj's thoughtful feedback regarding the novelty. We would like to respectfully clarify our perspective on this matter.
>
> Our primary novelty lies in the introduction of global autocorrelation as a foundational element for long-term prediction. Using global autocorrelation in long-term forecasting, to the best of our knowledge, has not been previously explored. While we acknowledge the potential for future refinement and the introduction of more intricate mathematical models, we believe the core novelty of our work is in emphasizing the significance of autocorrelation-based contrastive learning, AutoCon, within the sliding window-based long-term forecasting.
>
> To incorporate AutoCon, we have renovated a decomposition architecture that is suited for representation learning. While our model can be viewed as a combination of existing components, its architecture is specifically designed to emphasize the significance of the long-term branch. As we mentioned in Section 3.2, this aspect has often been overlooked by even complex existing models, yet our approach maintains simplicity while being effectively structured for long-term representation.
>
> We understand that perspectives on novelty may vary, and we respect alternative opinions. However, it is our strong belief that the incorporation of autocorrelation-based contrastive learning into the long-term forecasting framework, as demonstrated in our work, constitutes a meaningful advancement in the field.
>
> ### W2: Dataset Statistics and Extension to imbalanced data
>
> * 1: Dataset Statistics
> For reviewer pLfj's valuable suggestions, we added M-Table 5 summarizing dataset statistics in Appendix A.3. The dataset statistics will provide readers with a quick reference to understand the diverse nature of the datasets used in our experiments.
>
> * 2: Extension to imbalanced data
> Regarding reviewer pLfj's query on the performance of our methodology in imbalanced data scenarios, we appreciate the opportunity to clarify this aspect. As our focus is on time series forecasting, which is typically treated as a regression task, the concept of data imbalance manifests differently compared to classification tasks. In time series forecasting, data imbalance might refer to scenarios where certain patterns or trends are underrepresented. While our current study does not explicitly address this form of imbalance, our methodology's ability to learn long-term dependencies can be indirectly beneficial in scenarios where certain long-term trends and patterns are less frequent compared to repeated short-term fluctuations within the window. Specifically, AutoCon captures the overall tendency of the time series data across its entire length using the global autocorrelation.
> By considering the global structure of the data, it helps the model to identify and learn from long-term patterns, including those that may not occur frequently.
> Thus, we believe that in imbalanced scenarios, the model can recognize and adapt to these infrequent but potentially significant patterns, leading to more accurate forecasting. We acknowledge the importance of this aspect and will consider investigating this in future work to further enhance the robustness of our forecasting approach. Once again, we thank the reviewer for the constructive feedback.

---

> ### Author Response · Authors · 2023-11-20
>
> ### W3: Results on other evaluation metrics
> We concur with reviewer pLfj's recommendation to incorporate additional evaluation metrics, as this will further substantiate the superior performance of our method in comparison to the baselines.
> While existing metrics (*i.e.,* MSE and MAE) are standard metrics for long-term forecasting evaluation, they have limitations. Specifically, they may not adequately capture aspects such as the shape and temporal alignment of the time series, which are crucial for a comprehensive evaluation of a forecasting model's performance.
>
> To address these limitations, we introduced two additional metrics based on Dynamic Time Warping (DTW) [[1]](https://ieeexplore.ieee.org/abstract/document/1163055?casa_token=2XD8Dcxt2fIAAAAA:qR84AD50aWQv7paoubI5peCAiDRousaLO8n6egWNLkc66OF6IbbH9rx1cpvxs5HtveC3ZaB9NQ): Shape DTW and Temporal DTW [[2]](https://arxiv.org/abs/1909.09020). Shape DTW focuses on the similarity of the pattern or shape of the predicted sequence to the actual sequence, providing insight into the model's ability to capture the underlying pattern of the time series. Temporal DTW evaluates the alignment of the predicted sequence with the actual sequence, highlighting the model's accuracy in forecasting the timing of events.
>
> These additional metrics offer a more nuanced assessment of our model's performance, particularly in areas where MSE and MAE may fall short. Lower values in both Shape DTW and Temporal DTW indicate better performance, signifying lesser distortion between the predicted and actual sequences. As shown in R-Table 1, our method demonstrates superior performance for three datasets, which show obvious long-term variations, not only in MSE and MAE but also in these shape and temporal alignment-focused metrics.
> Although it is difficult to measure Accuracy or AUROC due to the absence of labels in the forecasting task, our comprehensive evaluation of the newly used metrics underscores the superiority of our method. Also, these experiments were added to Appendix B.6  in the revised version of the paper.
>
> **R-Table 1: Univariate forecasting results, evaluated using Shape and Temporal DTW in a 720-output setting across three datasets**
>
> | Dataset            | ETTh1            |                 | ETTh2        |                 | Electricity |  |
> |--------------------|------------------|-----------------|--------------|-----------------|----------------------|--------------------------|
> | **Model \ Metric** | Shape DTW        | Temporal DTW    | Shape DTW    | Temporal DTW    | Shape DTW | Temporal DTW |
> | **Ours**           | **17.14±1.653** | **59.66±1.739** | **42.38±0.630** | **13.47±1.793** | **80.73±7.7982**    | **0.09±0.014**           |
> | TimesNet           | 25.80±4.349      | 86.86±15.509    | 62.58±5.390  | 51.19±21.834    | 139.83±16.516      | 0.49±0.826               |
> | PatchTST           | 22.21±1.226      | 72.23±4.411     | 65.45±2.976  | 15.23±0.882     | 116.50±29.878      | 0.75±1.674               |
> | MICN               | 37.08±12.393     | 65.70±3.588     | 67.69±8.796  | 22.67±3.168     | 123.93±17.543      | 1.90±1.176               |
> | DLinear            | 58.32±1.955      | 155.21±8.587    | 82.53±3.627  | 24.88±2.403     | 88.70±3.550        | 0.18±0.145
>
> ### W4: Computational Cost
> In Section 4.4, we have briefly provided an analysis of the computational costs for each model. However, due to space limits, detailed information can be found in Appendix B.4.
> We measured the computational cost of various baselines for four different output lengths, ranging from 96 to 2160. In summary, our model recorded significantly lower inference times compared to the baselines except for linear models in a single inference process. Additionally, it demonstrated much faster inference speed compared to CNN-based, TimesNet, and Transformer-based models even as the prediction length increased.
> We would sincerely appreciate it if the reviewer could refer to it once more.
>
> ### Q1: Open-sourced code
> We have already attached our code as supplementary material in the open review system. If this is not made public after the review process, we are also willing to publicly open our code on Github and include the GitHub repository URL in the main paper as either an abstract or a footnote.
>
> ### References
> [[1]](https://ieeexplore.ieee.org/abstract/document/1163055?casa_token=2XD8Dcxt2fIAAAAA:qR84AD50aWQv7paoubI5peCAiDRousaLO8n6egWNLkc66OF6IbbH9rx1cpvxs5HtveC3ZaB9NQ) Dynamic programming algorithm optimization for spoken word recognition, IEEE transactions on acoustics, speech, and signal processing'1978
> [[2]](https://arxiv.org/abs/1909.09020) Shape and Time Distortion Loss for Training Deep Time Series Forecasting Models, NeurIPS'19

---

### Author Response · Authors · 2023-11-20
**Common response for all reviewers**

We sincerely appreciate all reviewers for their thoughtful comments and feedback. We carefully read all comments, and we tried to show as many empirical results as possible in response to the fruitful comments made by reviewers. For additional experiments, M-Table # and R-Table # (or M-Figure # and R-Figure #) indicate tables in Manuscripts and in Response, respectively. We also added several figures, which are provided in anonymous URLs, to the response for better understanding.

We actively embraced the feedback from the reviewers, and the relevant experiments were faithfully incorporated into the revised paper. We first provide a brief list of the changes made during this revision period.
1. We added dataset statistics at M-Table 5 in Appendix A.3 (R1)
2. We added two additional metrics at M-Table 8 in Appendix B.6 (R1)
3. We added detailed Figures related to M-Figure 2 at M-Figure 16 and 17 in Appendix C.1 (R2)
4. We added additional ablation results over two datasets at M-Figure 12 and 13 in Appendix B.2 (R2)
5. We added two representation baselines at M-Table 9 in Appendix C.3 (R3)
6. We added training pseudo-code in Appendix A.1 (R3)
7. We added ablation results for the long-term branch at M-Table 6 and 7 in Appendix B.3 (R4)
* R1, R2, R3, and R4 indicate Reviewer pLfj, Reviewer eBaK, Reviewer MDze, and Reviewer hiP7, respectively.

We appreciate the positive scores given by all the reviewers for soundness and presentation. We hope that the concerns raised by the reviewers will be addressed through our responses. If there is any misunderstanding or further questions, we are open to discussing them at any time.

---

### Meta-Review · Area_Chair_3NQP · 2023-12-06

**Metareview:**

This paper proposes a method based on contrastive learning to overcome the challenges of long-term forecasting. The authors have correctly noticed the need for capturing global autocorrelation in the time series and proposed using contrastive learning to achieve this. Comprehensive experiments are reported with mostly strong results. In response to the questions and concerns raised by the reviewers, the authors tried their best to answer them thoroughly and also conducted additional experiments to answer some of the questions. The authors are highly recommended to further revise their paper to address the outstanding issues before publication.

**Justification For Why Not Higher Score:**

It is not good enough to be accepted as a spotlight paper.

**Justification For Why Not Lower Score:**

With the revisions made in response to the reviewer comments, I think the paper has exceeded the acceptance standard of ICLR. Nevertheless, if there are many stronger papers, it is fine with me not to accept it.

---

### Decision · Program_Chairs · 2024-01-16

Accept (poster)